# Statistically Valid Variable Importance Assessment through Conditional Permutations

**Ahmad Chamma**
Inria, Universite Paris Saclay, CEA
`ahmad.chamma@inria.fr`

**Denis A. Engemann**
Roche Pharma Research and Early Development,
Neuroscience and Rare Diseases,
Roche Innovation Center Basel,
F. Hoffmann–La Roche Ltd., Basel, Switzerland
`denis.engemann@roche.com`

**Bertrand Thirion**
Inria, Universite Paris Saclay, CEA
`bertrand.thirion@inria.fr`

## Abstract

Variable importance assessment has become a crucial step in machine-learning applications when using complex learners, such as deep neural networks, on large-scale data. Removal-based importance assessment is currently the reference approach, particularly when statistical guarantees are sought to justify variable inclusion. It is often implemented with variable permutation schemes. On the flip side, these approaches risk misidentifying unimportant variables as important in the presence of correlations among covariates. Here we develop a systematic approach for studying Conditional Permutation Importance (CPI) that is model agnostic and computationally lean, as well as reusable benchmarks of state-of-the-art variable importance estimators. We show theoretically and empirically that *CPI* overcomes the limitations of standard permutation importance by providing accurate type-I error control. When used with a deep neural network, *CPI* consistently showed top accuracy across benchmarks. An experiment on real-world data analysis in a large-scale medical dataset showed that *CPI* provides a more parsimonious selection of statistically significant variables. Our results suggest that *CPI* can be readily used as drop-in replacement for permutation-based methods.

## 1   Introduction

Machine learning is an area of growing interest for biomedical research [Iniesta et al., 2016, Taylor and Tibshirani, 2015, Malley et al., 2011] for predicting biomedical outcomes from heterogeneous inputs [Hung et al., 2020, Zheng and Agresti, 2000, Giorgio et al., 2022, Sechidis et al., 2021]. Biomarker development is increasingly focusing on multimodal data including brain images, genetics, biological specimens and behavioral data [Coravos et al., 2019, Siebert, 2011, Ye et al., 2008, Castillo-Barnes et al., 2018, Yang et al., 2022]. Such high-dimensional settings with correlated inputs put strong pressure on model identification. With complex, often nonlinear models, it becomes harder to assess the role of features in the prediction, aka *variable importance* [Casalicchio et al., 2019, Altmann et al., 2010]. In epidemiological and clinical studies, one is interested in *population-level* feature importance, as opposed to instance-level feature importance.

In that context, variable importance is understood as *conditional* importance, meaning that it measures the information carried by one variable on the outcome *given* the others, as opposed to the easily accessible marginal importance of the variables. Conditional importance is necessary e.g. to assess

37th Conference on Neural Information Processing Systems (NeurIPS 2023).

whether a given measurement is worth acquiring, on top of others, for a diagnostic or prognostic task. As the identification of relevant variables is model-dependent and potentially unstable, point estimates of variable importance are misleading. One needs confidence intervals of importance estimates or statistical guarantees, such as type-I error control, i.e. the percentage of non-relevant variables detected as relevant (false positives). This control depends on the accuracy of the p-values on variable importance being non-zero [Cribbie, 2000].

Within the family of removal-based importance assessment methods [Covert et al., 2022], a popular model-agnostic approach is *permutation* variable importance, that measures the impact of shuffling a given variable on the prediction [Janitza et al., 2018]. By repeating the *permutation* importance analysis on permuted replicas of the variable of interest, importance values can be tested against the null hypothesis of being zero, yielding p-values that are valid under general distribution assumptions. Yet, statistical guarantees for permutation importance assessment do not hold in the presence of correlated variables, leading to selection of unimportant variables [Molnar et al., 2021, Hooker et al., 2021, Nicodemus et al., 2010, Stigler, 2005]. For instance, the method proposed in [Mi et al., 2021] is a powerful variable importance evaluation scheme, but it does not control the rate of type-I error.

In this work, we propose a general methodology for studying the properties of Conditional Permutation Importance in biomedical applications alongside tools for benchmarking variable importance estimators:

- Building on the previous literature on CPI, we develop theoretical results for the limitations regarding Permutation Importance (PI) and advantages of conditional Permutation Importance (CPI) given correlated inputs (section 3).

- We propose a novel implementation for CPI allowing us to combine the potential advantages of highly expressive base learners for prediction (a deep neural network) and a comparably lean Random Forest model as a conditional probability learner (section 4).

- We conduct extensive benchmarks on synthetic and heterogeneous multimodal real-world biomedical data tapping into different correlation levels and data-generating scenarios for both classification and regression (section 5).

- We propose a reusable library for simulation experiments and real-world applications of our method on a public GitHub repo `https://github.com/achamma723/Variable_Importance`.

## 2   Related work

A popular approach to interpret black-box predictive models is based on *locally interpretable*, i.e. *instance-based*, models. *LIME* [Ribeiro et al., 2016] provides local interpretable model-agnostic explanations by locally approximating a given complex model with a linear model around the instance of interest. *SHAP* [Burzykowski, 2020] is a popular package that measures *local* feature effects using the Shapley values from coalitional game theory.

However, global, i.e. population-level, explanations are better suited than instance-level explanations for epidemiological studies and scientific discovery in general. Many methods can be subsumed under the general category of removal-based approaches [Covert et al., 2022]. *Permutation* importance is defined as the decrease in a model score when the values of a single feature are randomly shuffled [Breiman, 2001]. This procedure breaks the relationship between the feature and the outcome, thus the drop in model performance expresses the relevance of the feature. Janitza et al. [2018] use an ensemble of Random Forests with the sample space equally partitioned. They approximate the null distribution based on the observed importance scores to provide p-values. Yet, this coarse estimate of the null distribution can give unstable results. Recently, a generic approach has been proposed in [Williamson et al., 2021] that measures the loss difference between models that include or exclude a given variable, also applied with LOCO (Leave One Covariate Out) in the work by Lei et al. [2018]. They show the asymptotic consistency of the model. However, their approach is intractable, given that it requires refitting the model for each variable. A simplified version has been proposed by Gao et al. [2022]. However, relying on linear approximations, some statistical guarantees from [Williamson et al., 2021] are potentially lost.

Another recent paper by Mi et al. [2021] has introduced model-agnostic explanation for black-box models based on the *permutation* approach. *Permutation* importance [Breiman, 2001] can work with

any learner. Moreover, it relies on a single model fit, hence it is an efficient procedure. Strobl et al. [2008] pointed out limitations with the *permutation* approach in the face of correlated variables. As an alternative, they propose a *conditional permutation* importance by shuffling the variable of interest conditionally on the other variables. However, the solution was specific to Random Forests, as it is based on bisecting the space with the cutpoints extracted during the building process of the forest.

With the *Conditional Randomization Test* proposed by Candes et al. [2017], the association between the outcome $y$ and the variable of interest $x^j$ conditioned on $\mathbf{x}^{-\mathbf{j}}$ is estimated. The variable of interest is sampled conditionally on the other covariates multiple times to compute a test statistic and p-values. However, this solution is limited to generalized linear models and is computationally expensive. Finally, a recent paper by [Watson and Wright, 2021] showed the necessity of conditional schemes and introduced a knockoff sampling scheme, whereby the variable of interest is replaced by its knockoff to monitor any drop in performance of the leaner used without refitting. This method is computationally inexpensive, and enjoys statistical guarantees from from [Lei et al., 2018]. However, it depends on the quality of the knockoff sampling where even a relatively small distribution shift in knockoff generation can lead to large errors at inference time.

Other work has presented comparisons of select models within distinct communities [Liu et al., 2021, Chipman et al., 2010, Janitza et al., 2018, Mi et al., 2021, Altenmüller et al., 2021], however, lacking conceptualization from a unified perspective. In summary, previous work has established potential advantages of conditional permutation schemes for inference of variable importance. Yet, the lack of computationally scalable approaches has hampered systematic investigations of different permutation schemes and their comparison with alternative techniques across a broader range of predictive modeling settings.

# 3 Permutation importance and its limitations

## 3.1 Preliminaries

**Notations**    We will use the following system of notations. We denote matrices, vectors, scalar variables and sets by bold uppercase letters, bold lowercase letters, script lowercase letters, and calligraphic letters, respectively (e.g. $\mathbf{X}$, $\mathbf{x}$, $x$, $\mathcal{X}$). We call $\mu$ the function that maps the sample space $\mathcal{X} \subset \mathbb{R}^p$ to the sample space $\mathcal{Y} \subset \mathbb{R}$ and $\hat{\mu}$ is an estimate of $\mu$. Permutation procedures will be represented by (*perm*). We denote by $[\![n]\!]$ the set $\{1, \ldots, n\}$.

Let $\mathbf{X} \in \mathbb{R}^{n \times p}$ be a design matrix where the i-th row and the j-th column are denoted $\mathbf{x_i}$ and $\mathbf{x^j}$ respectively. Let $\mathbf{X^{-j}} = (\mathbf{x^1}, \ldots, \mathbf{x^{j-1}}, \mathbf{x^{j+1}}, \ldots, \mathbf{x^P})$ be the design matrix, where the $j^{th}$ column is removed, and $\mathbf{X^{(j)}} = (\mathbf{x^1}, \ldots, \mathbf{x^{j-1}}, \{\mathbf{x^j}\}^{perm}, \mathbf{x^{j+1}}, \ldots, \mathbf{x^P})$ the design matrix with the $j^{th}$ column shuffled. The rows of $\mathbf{X^{-j}}$ and $\mathbf{X^{(j)}}$ are denoted $\mathbf{x_i^{-j}}$ and $\mathbf{x_i^{(j)}}$ respectively, for i $\in [\![n]\!]$.

**Problem setting**    Machine learning inputs are a design matrix $\mathbf{X}$ and a target $\mathbf{y} \in \mathbb{R}^n$ or $\in \{0, 1\}^n$ depending on whether it is a regression or a classification problem. Throughout the paper, we rely on an i.i.d. sampling train / test partition scheme where the $n$ samples are divided into $n_{train}$ training and $n_{test}$ test samples and consider that $\mathbf{X}$ and $\mathbf{y}$ are restricted to the test samples - the training samples were used to obtain $\hat{\mu}$.

## 3.2 The *permutation* approach leads to false detections in the presence of correlations

A known problem with *permutation* variable importance is that if features are correlated, their importance is typically over-estimated [Strobl et al., 2008], leading to a loss of type-I error control. However, this loss has not been precisely characterized yet, which we will work through for the linear case. We use the setting of [Mi et al., 2021], where the estimator $\hat{\mu}$, computed with empirical risk minimization under the training set, is used to assess variable importance on a new set of data (test set). We consider a regression model with a least-square loss function for simplicity. The importance of variable $\mathbf{x^j}$ is computed as follows:

$$\hat{m}^j = \frac{1}{n_{test}} \sum_{i=1}^{n_{test}} \left( (y_i - \hat{\mu}(\mathbf{x_i^{(j)}}))^2 - (y_i - \hat{\mu}(\mathbf{x_i}))^2 \right). \tag{1}$$

Let $\varepsilon_i = y_i - \mu(\mathbf{x_i})$ for $i \in [\![n_{test}]\!]$. Re-arranging terms yields

$$\hat{m}^j = \frac{1}{n_{test}} \sum_{i=1}^{n_{test}} (\hat{\mu}(\mathbf{x_i}) - \hat{\mu}(\mathbf{x_i^{(j)}}))(2\mu(\mathbf{x_i}) - \hat{\mu}(\mathbf{x_i}) - \hat{\mu}(\mathbf{x_i^{(j)}})) + 2\varepsilon_i). \qquad (2)$$

Mi et al. [2021] argued that these terms vanish when $n_{test} \to \infty$. But it is not the case as long as the training set is fixed. In order to get tractable computation, we assume that $\mu$ and $\hat{\mu}$ are linear functions: $\mu(\mathbf{x}) = \mathbf{x}\mathbf{w}$ and $\hat{\mu}(\mathbf{x}) = \mathbf{x}\hat{\mathbf{w}}$. Let us further consider that $\mathbf{x^j}$ is a null feature, i.e. $w^j = 0$. This yields $\mathbf{x}\mathbf{w} = x^j w^j + \mathbf{x^{-j}}\mathbf{w^{-j}} = \mathbf{x^{-j}}\mathbf{w^{-j}}$. Denoting the standard dot product by $\langle .,. \rangle$, this leads to (Detailed proof of getting from Eq. 2 to Eq. 3 can be found in supplement section A)

$$\hat{m}^j = \frac{2\hat{w}^j}{n_{test}} \left\langle \mathbf{x^j} - \{\mathbf{x^j}\}^{perm}, \mathbf{X^{-j}}(\mathbf{w^{-j}} - \hat{\mathbf{w}}^{-j}) + \varepsilon \right\rangle \qquad (3)$$

as $(\|\mathbf{x^j}\|^2 - \|\{\mathbf{x^j}\}^{perm}\|^2) = 0$. Next, $\frac{1}{n_{test}}\langle\{\mathbf{x^j}\}^{perm}, \mathbf{X^{-j}}(\mathbf{w^{-j}} - \hat{\mathbf{w}}^{-j})\rangle \to 0$ and $\frac{1}{n_{test}}\langle \mathbf{x^j} - \{\mathbf{x^j}\}^{perm}, \varepsilon \rangle \to 0$ when $n_{test} \to \infty$ with speed $\frac{1}{\sqrt{n_{test}}}$ from the Berry-Essen theorem, assuming that the first three moments of these quantities are bounded and that the test samples are i.i.d. Let us assume that the correlation within $\mathbf{X}$ takes the following form: $\mathbf{x^j} = \mathbf{X^{-j}}\mathbf{u} + \boldsymbol{\delta}$, where $\mathbf{u} \in \mathbb{R}^{p-1}$ and $\boldsymbol{\delta}$ is a random vector independent of $\mathbf{X^{-j}}$. By contrast, $\frac{2\hat{w}^j}{n_{test}}\langle \mathbf{x^j}, \mathbf{X^{-j}}(\mathbf{w^{-j}} - \hat{\mathbf{w}}^{-j})\rangle$ has a non-zero limit $2\hat{w}^j \mathbf{u}^T Cov(\mathbf{X^{-j}})(\mathbf{w^{-j}} - \hat{\mathbf{w}}^{-j})$, where $Cov(\mathbf{X^{-j}}) = \lim_{n_{test}\to\infty}\frac{\mathbf{X^{-j}}^T\mathbf{X^{-j}}}{n_{test}}$ (remember that both $\mathbf{w^{-j}}$ and $\hat{\mathbf{w}}^{-j}$ are fixed, because the training set is fixed). Thus, the permutation importance of a null but correlated variable does not vanish when $n_{test} \to \infty$, implying that this inference scheme will lead to false positives.

## 4 *Conditional sampling*-based feature importance

### 4.1 Main result

We define the permutation of variable $x^j$ conditional to $\mathbf{x^{-j}}$, as a variable $\tilde{x}^j$ that retains the dependency of $x^j$ with respect to the other variables in $\mathbf{x^{-j}}$, but where the independent part is shuffled; $\tilde{\mathbf{x}}^{(j)}$ is the vector $\mathbf{x}$ where $x^j$ is replaced by $\tilde{x}^j$. We propose two constructions below (see Fig. E1). In the case of regression, this leads to the following importance estimator:

$$\hat{m}^j_{CPI} = \frac{1}{n_{test}} \sum_{i=1}^{n_{test}} \left( (y_i - \hat{\mu}(\tilde{\mathbf{x}}_\mathbf{i}^{(j)}))^2 - (y_i - \hat{\mu}(\mathbf{x_i}))^2 \right). \qquad (4)$$

As noted by Watson and Wright [2021], this inference is correct, as in traditional permutation tests, as long as one wishes to perform inference conditional to $\hat{\mu}$. However, the following proposition states that this inference has much wider validity in the asymptotic regime.

**Proposition.** *Assuming that the estimator $\hat{\mu}$ is obtained from a class of functions $\mathcal{F}$ with sufficient regularity, i.e. that it meets conditions (A1, A2, A3, A4, B1 and B2) defined in supplementary material, the importance score $\hat{m}^j_{CPI}$ defined in (4) cancels when $n_{train} \to \infty$ and $n_{test} \to \infty$ under the null hypothesis, i.e. the j-th variable is not significant for the prediction. Moreover, the Wald statistic $z^j = \frac{mean(\hat{m}^j_{CPI})}{std(\hat{m}^j_{CPI})}$ obtained by dividing the mean of the importance score by its standard deviation asymptotically follows a standard normal distribution.*

This implies that in the large sample limit, the p-value associated with $z^j$ controls the type-I error rate for all optimal estimators in $\mathcal{F}$.

The proof of the proposition is given in the supplement (section C). It consists in observing that the importance score defined in (4) is 0 for the class of learners discussed in [Williamson et al., 2021], namely those that meet a certain set of convergence guarantees and are invariant to arbitrary change of their $j^{th}$ argument, conditional on the others. In the supplement, we also restate the precise technical conditions under which the importance score $\hat{m}^j_{CPI}$ used is (asymptotically) valid, i.e. leads to a Wald-type statistic that behaves as a standard normal under the null hypothesis.

It is easy to see that for the setting in Sec. 3.2, all terms in Eq. 4 vanish with speed $\frac{1}{\sqrt{n_{test}}}$.

## 4.2 Practical estimation

Next, we present algorithms for computing conditional permutation importance. We propose two constructions for $\tilde{x}^j$, the conditionally permuted counterpart of $x^j$. The first one is additive: on test samples, $x^j$ is divided into the predictable and random parts $\tilde{x}^j = \mathbb{E}(x^j | \mathbf{x}^{-\mathbf{j}}) + \left(x^j - \mathbb{E}(x^j | \mathbf{x}^{-\mathbf{j}})\right)^{perm}$, where the residuals of the regression of $x^j$ on $\mathbf{x}^{-\mathbf{j}}$ are shuffled to obtain $\tilde{x}^j$. In practice, the expectation is obtained by a universal but efficient estimator, such as a random forest trained on the test set.

The other possibility consists in using a random forest (RF) model to fit $x^j$ from $\mathbf{x}^{-\mathbf{j}}$ and then sample the prediction within leaves of the RF.

Random shuffling is applied B times. For instance, using the additive construction, a shuffling of the residuals $\tilde{e}^{\mathbf{j},\mathbf{b}}$ for a given $b \in [\![B]\!]$ allows to reconstruct the variable of interest as the sum of the predicted version and the shuffled residuals, that is

$$\tilde{\mathbf{x}}^{\mathbf{j},\mathbf{b}} = \hat{\mathbf{x}}^{\mathbf{j}} + \tilde{e}^{\mathbf{j},\mathbf{b}}. \tag{5}$$

Let $\tilde{\mathbf{X}}^{\mathbf{j},\mathbf{b}} = (\mathbf{x}^{\mathbf{1}}, \ldots, \mathbf{x}^{\mathbf{j-1}}, \tilde{\mathbf{x}}^{\mathbf{j},\mathbf{b}}, \mathbf{x}^{\mathbf{j+1}}, \ldots, \mathbf{x}^{\mathbf{P}}) \in \mathbb{R}^{n_{test} \times p}$ be the new design matrix including the reconstructed version of the variable of interest $\mathbf{x}^{\mathbf{j}}$. Both $\tilde{\mathbf{X}}^{\mathbf{j},\mathbf{b}}$ and the target vector $\mathbf{y}$ are fed to the loss function in order to compute a loss score $l_i^{j,b} \in \mathbb{R}$ defined by

$$l_i^{j,b} = \begin{cases} y_i \log \left( \frac{S(\hat{y}_i)}{S(\tilde{y}_i^b)} \right) + (1 - y_i) \log \left( \frac{1 - S(\hat{y}_i)}{1 - S(\tilde{y}_i^b)} \right) \\ (y_i - \tilde{y}_i^b)^2 - (y_i - \hat{y}_i)^2 \end{cases} \tag{6}$$

for binary and regression cases respectively where $i \in [\![n_{test}]\!]$, $j \in [\![p]\!]$, $b \in [\![B]\!]$, $i$ indexes a test sample of the dataset, $\hat{y}_i = \hat{\mu}(\mathbf{x_i})$ and $\tilde{y}_i^b = \hat{\mu}(\tilde{\mathbf{x}}_{\mathbf{i}}^{\mathbf{j},\mathbf{b}})$ is the new fitted value following the reconstruction of the variable of interest with the $b^{th}$ residual shuffled and $S(x) = \frac{1}{1+e^{-x}}$.

The variable importance scores are computed as the double average over the number of permutations $B$ and the number of test samples $n_{test}$ (line 15 of Alg. 1), while their standard deviations are computed as the square root of the average over the test samples of the quadratic deviation over the number of permutations (line 17). Note that, unlike Williamson et al. [2021], the variance estimator is non-vanishing, and thus can be used as a plugin. A $z_{CPI}^j$ statistic is then computed by dividing the mean of the corresponding importance scores with the corresponding standard deviation (line 18). P-values are computed using the cumulative distribution function of the standard normal distribution (line 19). The conditional sampling and inference steps are summarized in Algorithm 1. This leads to the *CPI-DNN* method when $\hat{\mu}$ is a deep neural network, or *CPI-RF* when $\hat{\mu}$ is a random forest. Supplementary analysis reporting the computational advantage of *CPI-DNN* over a remove-and-relearn alternative a.k.a. *LOCO-DNN*, can be found in supplement (section D), which justifies its *computational leanness*.

# 5 Experiments & Results

In all experiments, we refer to the original implementation of the different methods in order to maintain a fair comparison. Regarding *Permfit-DNN, CPI-DNN* and *CPI-RF* models specifically, our implementation involves a 2-fold internal validation (the training set of further split to get validation set for hyperparameter tuning). The scores from different splits are thus concatenated to compute the final variable importance. We focus on the *Permfit-DNN* and *CPI-DNN* importance estimators that use a deep neural network as learner $\hat{\mu}$, using standard permutation and algorithm 1, respectively. All experiments are performed with 100 runs. The evaluation metrics are detailed in the supplement (section E).

## 5.1 Experiment 1: Type-I error control and accuracy when increasing variable correlation

We compare the performance of *CPI-DNN* with that of *Permfit-DNN* by applying both methods across different correlation scenarios. The data $\{\mathbf{x_i}\}_{i=1}^n$ follow a Gaussian distribution with a prescribed covariance structure $\mathbf{\Sigma}$ i.e. $\mathbf{x_i} \sim \mathcal{N}(0, \mathbf{\Sigma}) \forall i \in [\![n]\!]$. We consider a block-designed covariance matrix $\mathbf{\Sigma}$ of 10 blocks with an equal correlation coefficient $\rho \in \{0, 0.2, 0.5, 0.8\}$ among the variables of

**Algorithm 1 Conditional sampling step**: The algorithm implements the conditional sampling step in place of the permutation approach when computing the p-value of variable $x^j$

---

**Require:** $\mathbf{X} \in \mathbb{R}^{n_{test} \times p}$, $\mathbf{y} \in \mathbb{R}^{n_{test}}$, $\hat{\mu}$: estimator, $l$: loss function, $\mathrm{RF}_j$: learner trained to predict $x^j$ from $\mathbf{x^{-j}}$

1: $B \leftarrow$ number of permutations
2: $\mathbf{X^{-j}} \leftarrow \mathbf{X}$ with j-th column removed
3: **for** i = 1 to $n_{test}$ **do**
4:     $\hat{x}_i^j \leftarrow \mathrm{RF}_j(\mathbf{x_i^{-j}})$
5: **end for**
6: Residuals $\epsilon^{\mathbf{j}} \leftarrow \mathbf{x^j} - \hat{\mathbf{x}}^{\mathbf{j}}$
7: **for** b = 1 to B **do**
8:     $\tilde{\epsilon}^{\mathbf{j},\mathbf{b}} \leftarrow$ Random Shuffling($\epsilon^{\mathbf{j}}$)
9:     $\tilde{\mathbf{x}}^{\mathbf{j},\mathbf{b}} \leftarrow \hat{\mathbf{x}}^{\mathbf{j}} + \tilde{\epsilon}^{\mathbf{j},\mathbf{b}}$
10:     **for** i = 1 to $n_{test}$ **do**
11:         $\tilde{y}_i^b \leftarrow \hat{\mu}(\tilde{\mathbf{x}}_{\mathbf{i}}^{\mathbf{j},\mathbf{b}})$
12:         compute $l_i^{j,b}$

13:     **end for**
14: **end for**
15: $\mathrm{mean}(\hat{m}_{CPI}^j) = \frac{1}{n_{test}} \frac{1}{B} \sum_{i=1}^{n_{test}} \sum_{b=1}^{B} l_i^{j,b}$
16: $\tau_i^j = \left( \frac{1}{B} \sum_{b=1}^{B} l_i^{j,b} - mean(\hat{m}_{CPI}^j) \right)^2$
17: $\mathrm{std}(\hat{m}_{CPI}^j) = \sqrt{\frac{1}{n_{test}-1} \sum_{i=1}^{n_{test}} \tau_i^j}$
18: $z_{CPI}^j = \frac{\mathrm{mean}(\hat{m}_{CPI}^j)}{\mathrm{std}(\hat{m}_{CPI}^j)}$
19: $p^j \leftarrow 1 - cdf(z_{CPI}^j)$

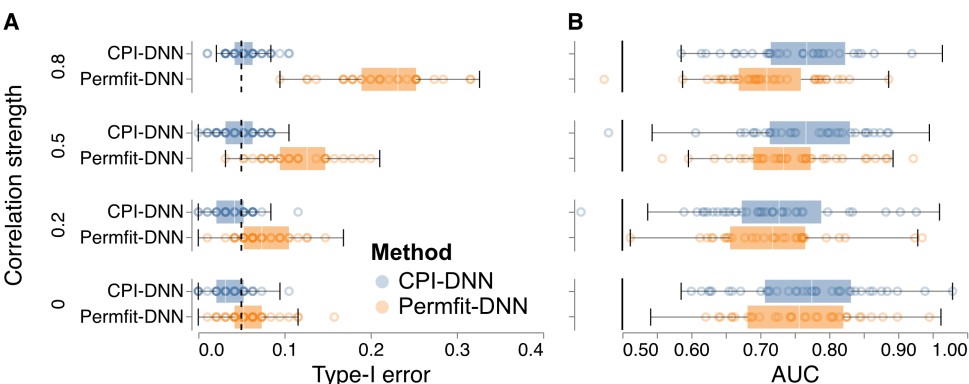

Figure 1: **CPI-DNN vs Permfit-DNN**: Performance at detecting important variables on simulated data with $n = 300$ and $p = 100$. **(A)**: The type-I error quantifies to which extent the rate of low p-values ($p < 0.05$) exceeds the nominal false positive rate. **(B)**: The AUC score measures to which extent variables are ranked consistently with the ground truth. Dashed line: targeted type-I error rate. Solid line: chance level.

each block. In this experiment, $p = 100$ and $n = 300$. The first variable of each of the first 5 blocks is chosen to predict the target $y$ with the following model, where $\epsilon \sim \mathcal{N}(0, \mathbf{I})$:

$$y_i = x_i^1 + 2 \log(1 + 2(x_i^{11})^2 + (x_i^{21} + 1)^2) + x_i^{31} x_i^{41} + \epsilon_i, \; \forall i \in [\![n]\!]$$

The AUC score and type-I error are presented in Fig. 1. Power and computation time are reported in the supplement Fig. 1 - S1. Based on the AUC scores, *Permfit-DNN* and *CPI-DNN* showed virtually identical performance. However, *Permfit-DNN* lost type-I error control when correlation in $\mathbf{X}$ is increased, while *CPI-DNN* always controlled the type-I error at the targeted rate.

### 5.2 Experiment 2: Performance across different settings

In the second setup, we check if *CPI-DNN* and *Permfit-DNN* control the type-I error with an increasing total number of samples $n$. The data are generated as previously, with a correlation $\rho = 0.8$. We fix the number of variables $p$ to 50 while the number of samples $n$ increases from 100 to 1000 with a step size of 100. We use 5 different models to generate the outcome $\mathbf{y}$ from $\mathbf{X}$: *classification*, *Plain linear*, *Regression with ReLu*, *Interactions only* and *Main effects with interactions*. Further details regarding each data-generating scenario can be found in supplement (section G).

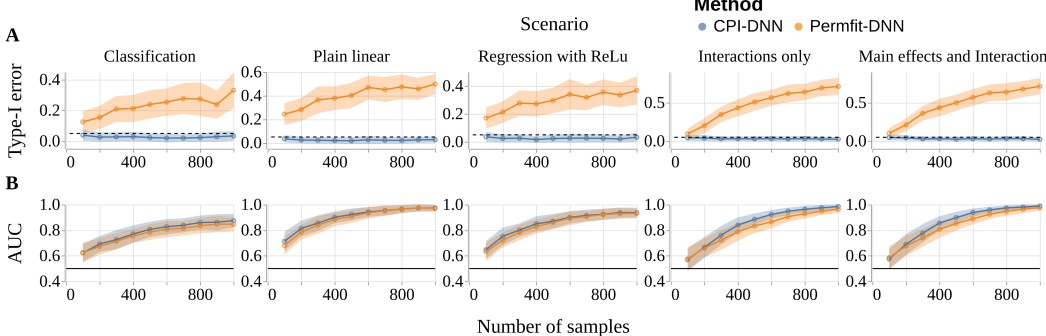

Figure 2: **Model comparisons across data-generating scenarios**: The **(A)** type-I error and **(B)** AUC scores of *Permfit-DNN* and *CPI-DNN* are plotted as function of sample size for five different settings. The number $n$ of samples increased from 100 to 1000 with a step size of 100. The number of variables $p$ was set to 50. Dashed line: targeted type-I error rate. Solid line: chance level.

The AUC score and type-I error of *Permfit-DNN* and *CPI-DNN* are shown as a function of sample size in Fig. 2. The accuracy of the two methods was similar across data-generating scenarios, with a slight reduction in the AUC scores of *Permfit-DNN* as compared to *CPI-DNN*. Only *CPI-DNN* controlled the rate of type-I error in the different scenarios at the specified level of 0.05. Thus, *CPI-DNN* provided an accurate ranking of the variables according to their importance score while, at the same time, controlling for the type-I error in all scenarios.

### 5.3 Experiment 3: Performance benchmark across methods

In the third setup, we include *Permfit-DNN* and *CPI-DNN* in a benchmark with other state-of-the-art methods for variable importance using the same setting as in Experiment 2, while fixing the total number of samples $n$ to 1000. We consider the following methods:

- Marginal Effects: A univariate linear model is fit to explain the response from each of the variables separately. The importance scores are then obtained from the ensuing p-values.
- Conditional-RF [Strobl et al., 2008]: A conditional variable importance approach based on a Random Forest model. This method provides p-values.
- $d_0$CRT [Liu et al., 2021, Nguyen et al., 2022]: The Conditional Randomization Test with distillation, using a sparse linear or logistic learner.
- Lazy VI [Gao et al., 2022].
- Permfit-DNN [Mi et al., 2021].
- LOCO [Lei et al., 2018]: This method applies the remove-and-retrain approach.
- cpi-knockoff [Watson and Wright, 2021]: Similar to CPI-RF, but permutation steps are replaced by a sampling step with a knockoff sampler.
- CPI-RF: This corresponds to the method in Alg. 1, where $\hat{\mu}$ is a Random Forest.
- CPI-DNN: This corresponds to the method in Alg. 1, where $\hat{\mu}$ is a DNN.

The extensive benchmarks on baselines and competing methods that provide p-values are presented in Fig. 3. For type-I error, $d_0$CRT, *CPI-RF*, *CPI-DNN*, *LOCO* and *cpi-knockoff* provided reliable control, whereas Marginal effects, *Permfit-DNN*, *Conditional-RF* and *Lazy VI* showed less consistent results across scenarios. For AUC, we observed that marginal effects performed poorly, as they do not use a proper predictive model. *LOCO* and *cpi-knockoff* behave similarly. $d_0$CRT performed well when the data-generating model was linear and did not include interaction effects. *Conditional-RF* and *CPI-RF* showed reasonable performance across scenarios. Finally, *Permfit-DNN* and *CPI-DNN* outperformed all the other methods, closely followed by *Lazy VI*.

Additional benchmarks on popular methods that do not provide p-values, e.g. BART [Chipman et al., 2010] or local and instance-based methods such as Shapley values [Kumar et al., 2020], are reported in the supplement (section H). The performance of these methods in terms of power and computation

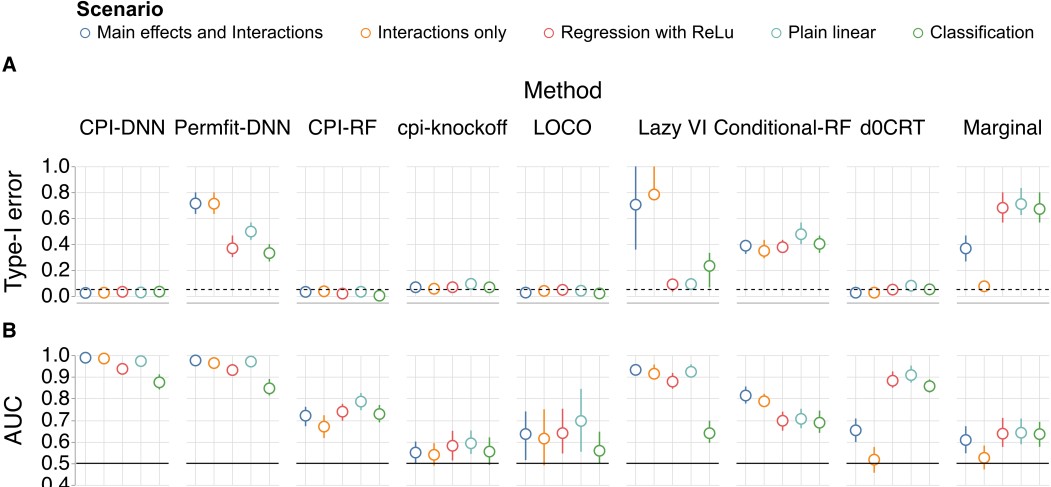

Figure 3: **Extended model comparisons**: *CPI-DNN* and *Permfit-DNN* were compared to baseline models (outer columns) and competing approaches across data-generating scenarios (inner columns). Prediction tasks were simulated with $n = 1000$ and $p = 50$. **(A)**: Type-I error. **(B)**: AUC scores. Dashed line: targeted type-I error rate. Solid line: chance level.

time are reported in the supplement Figs. 3 - S2 & 3 - S3 respectively. Additional inspection of power showed that across data generating scenarios, *CPI-DNN*, *Permfit-DNN* and *conditional-RF* showed strong results. *Marginal* and *d0CRT* performed only well in scenarios without interaction effects. *CPI-RF*, *cpi-knockoff*, *LOCO* and *Lazy VI* performed poorly. Finally, to put estimated variable importance in perspective with model capacity, we benchmarked prediction performance of the underlying learning algorithms in the supplement Fig. 3 - S4.

## 5.4   Experiment 4: *Permfit-DNN* vs *CPI-DNN* on Real Dataset UKBB

Large-scale simulations comparing the performance of *CPI-DNN* and *Permfit-DNN* are conducted in supplement (section L). We conducted an empirical study of variable importance in a biomedical application using the non-conditional permutation approach Permfit-DNN (no statistical guarantees for correlated inputs) and the safer CPI-DNN approach. A recent real-world data analysis of the UK Biobank dataset reported successful machine learning analysis of individual characteristics. The UK Biobank project (UKBB) curates phenotypic and imaging data from a prospective cohort of volunteers drawn from the general population of the UK [Constantinescu et al., 2022]. The data is provided by the UKBB operating within the terms of an Ethics and Governance Framework. The work focused on age, cognitive function and mood from brain images and social variables and put the ensuing models in relation to individual life-style choices regarding sleep, exercise, alcohol and tobacco [Dadi et al., 2021].

A coarse analysis of variable importance was presented, in which entire blocks of features were removed. It suggested that variables measuring brain structure or brain activity were less important for explaining the predictions of cognitive or mood outcomes than socio-demographic characteristics. On the other hand, brain imaging phenotypes were highly predictive of the age of a person, in line with the brain-age literature [Cole and Franke, 2017]. In this benchmark, we explored variable-level importance rankings provided by the *CPI-DNN* and *Permfit-DNN* methods.

The real-world empirical benchmarks on predicting personal characteristics and life-style are summarized in Fig. 4. Results in panel **(A)** suggest that highest agreement for rankings between *CPI-DNN* and *Permfit-DNN* was achieved for social variables (bottom left, orange squares). At the same time, *CPI-DNN* flagged more brain-related variables as relevant (bottom right, circles). We next computed counts and percentage and broke down results by variable domain (Fig. 4, **B**). Naturally, the total relevance for brain versus social variables varied by outcome. However, as a tendency, *CPI-DNN* seemed more selective as it flagged fewer variables as important (blue) beyond those flagged as important by both methods (light blue). This was more pronounced for social variables where *CPI-DNN*

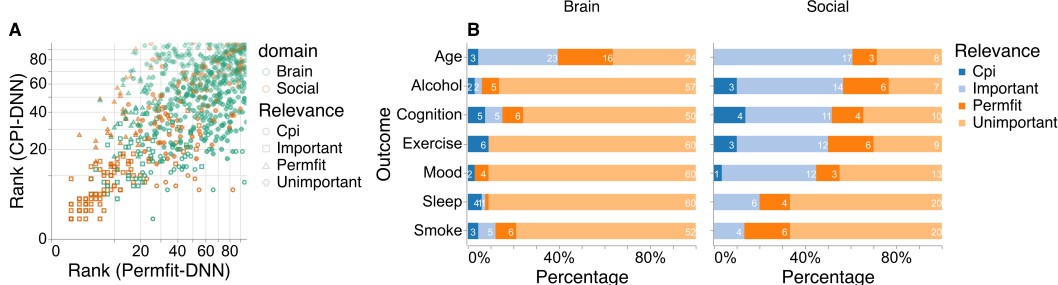

Figure 4: **Real-world empirical benchmark**: Prediction of personal characteristics (age, cognition, mood) and life-style habits (alcohol consumption, sleep, exercise & smoking) from various sociodemographic and brain-imaging derived phenotypes in a sample of $n = 8357$ volunteers from the UK Biobank. **(A)** plots variable rankings for *Permfit-DNN* (x axis) versus *CPI-DNN* (y axis) across all outcomes. Color: variable domain (brain versus social). Shape: variables classified by both methods as important (squares), unimportant (crosses) or by only one of the methods, *i.e.*, *CPI-DNN* (circles) or *Permfit-DNN* (triangles). **(B)** presents a detailed breakdown of percentage and counts of variable classification split by variable domain.

sometimes added no further variables. As expected by the impact of aging on brain structure and function, brain data was most important for age-prediction compared to other outcomes. Interestingly, most disagreements between the methods occurred in this setting as *CPI* rejected 16 out of 66 brain inputs that were found as important by *Permfit*. This outlines the importance of correlations between brain variables, that lead to spurious importance findings with *Permfit*. We further explored the utility of our approach for age-prediction from neuromagnetic recordings [Engemann et al., 2020] and observed that *CPI-DNN* readily selected relevant frequency bands without fine-tuning the approach (section M in the supplement).

## 6    Discussion

In this work, we have developed a framework for studying the behavior of marginal and conditional permutation methods and proposed the *CPI-DNN* method, that was inspired by the limitations of the *Permfit-DNN* approach. Both methods build on top of an expressive DNN learner, and both methods turned out superior to competing methods at detecting relevant variables, leading to high AUC scores across various simulated scenarios. However, our theoretical results predicted that *Permfit-DNN* would not control type-I error with correlated data, which was precisely what our simulation-based analyzes confirmed for different data-generating scenarios (Fig. 1 - 2). Other popular methods (Fig. 3) showed similar failures of type-I error control across scenarios or only worked well in a subset of tasks. Instead, *CPI-DNN* achieved control of type-I errors by upgrading the *permutation* to *conditional permutation*. The consequences were pronounced for correlated predictive features arising from generative models with product terms, which was visible even with a small fraction of data points for model training. Among alternatives, the *Lazy VI* approach [Gao et al., 2022] obtained an accuracy almost as good as *Permfit-DNN* and *CPI-DNN* but with an unreliable type-I error control.

Taken together, our results suggest that *CPI-DNN* may be a practical default choice for variable importance estimation in predictive modeling. A practical validation of the standard normal distribution assumption for the non important variables can be found in supplement (section N). The *CPI* approach is generic and can be implemented for any combination of learning algorithms as a base learner or conditional means estimator. *CPI-DNN* has a linear and quadratic complexity in the number of samples and variables, respectively. This is of concern when modeling the conditional distribution of the variable of interest which lends itself to high computational complexity. In our work, Random Forests proced to be useful default estimators as they are computationally lean and their model complexity, given reasonable default choices implemented in standard software, can be well controlled by tuning the tree depth. In fact, our supplementary analyses (section O) suggest that proper hyperparameter tuning was sufficient to obtain good calibration of p-values. As a potential limitation, it is noteworthy the current configuration of our approach uses a deep neural network as

the base learner. Therefore, in general, more samples might be needed for good model performance, hence, improved model interpretation.

Our real-world data analysis demonstrated that *CPI-DNN* is readily applicable, providing similar variable rankings as *Permfit-DNN*. The differences observed are hard to judge as the ground truth is not known in this setting. Moreover, accurate variable selection is important to obtain unbiased interpretations which are relevant for data-rich domains like econometrics, epidemiology, medicine, genetics or neuroscience. In that context, it is interesting that recent work raised doubts about the signal complexity in the UK biobank dataset [Schulz et al., 2020], which could mean that underlying predictive patterns are spread out over correlated variables. In the subset of the UK biobank that we analysed, most variables actually had low correlation values (Fig. E4), which would explain why *CPI-DNN* and *Permfit-DNN* showed similar results. Nevertheless, our empirical results seem compatible with our theoretical results as *CPI-DNN* flagged fewer variables as important, pointing at stricter control of type-I errors, which is a welcome property for biomarker discovery.

When considering two highly correlated variables $x_1$ and $x_2$, the corresponding conditional importance of both variables is 0. This problem is linked to the very definition of conditional importance, and not to the *CPI* procedure itself. The only workaround is to eliminate, prior to importance analysis, degenerate cases where conditional importance cannot be defined. Therefore, possible future directions include inference on groups of variables, e.g, gene pathways, brain regions, while preserving statistical control offered by *CPI-DNN*.

**Acknowledgement** This work has been supported by Bertrand Thirion and is supported by the KARAIB AI chair (ANR-20-CHIA-0025-01), and the H2020 Research Infrastructures Grant EBRAIN-Health 101058516. D.E. is a full-time employee of F. Hoffmann-La Roche Ltd.

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

## A  Supplement proof - getting from Eq. 2 to Eq. 3

$$
\hat{m}^j = \frac{1}{n_{test}} \sum_{i=1}^{n_{test}} (\hat{\mu}(\mathbf{x_i}) - \hat{\mu}(\mathbf{x_i^{(j)}}))(2\mu(\mathbf{x_i}) - \hat{\mu}(\mathbf{x_i}) - \hat{\mu}(\mathbf{x_i^{(j)}}) + 2\varepsilon_i) \, 2
$$

$$
= \frac{1}{n_{test}} \sum_{i=1}^{n_{test}} (\mathbf{x_i^{-j}}\hat{\mathbf{w}}^{-j} + x_i^j \hat{w}^j - \mathbf{x_i^{-j}}\hat{\mathbf{w}}^{-j} - \{x_i^j\}^{perm}\hat{w}^j)(2\mathbf{x_i^{-j}}\mathbf{w^{-j}} - 2\mathbf{x_i^{-j}}\hat{\mathbf{w}}^{-j} - (x_i^j \hat{w}^j + \{x_i^j\}^{perm}\hat{w}^j) + 2\varepsilon_i)
$$

$$
= \frac{2\hat{w}^j}{n_{test}} \sum_{i=1}^{n_{test}} (x_i^j - \{x_i^j\}^{perm})(\mathbf{x_i^{-j}}(\mathbf{w^{-j}} - \hat{\mathbf{w}}^{-j}) + \varepsilon_i) - \hat{w}^j((x_i^j)^2 - (\{x_i^j\}^{perm})^2)
$$

$$
= \frac{2\hat{w}^j}{n_{test}} \sum_{i=1}^{n_{test}} (x_i^j - \{x_i^j\}^{perm})(\mathbf{x_i^{-j}}(\mathbf{w^{-j}} - \hat{\mathbf{w}}^{-j}) + \varepsilon_i) \, 3
$$

## B  Diagram of CPI constructions

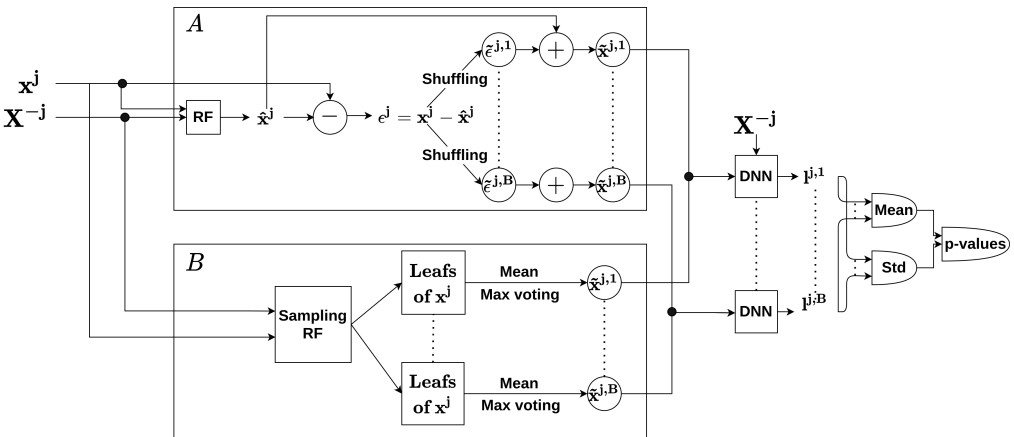

Figure E1: **CPI-DNN's constructions**: Constructing the variable of interest $\tilde{\mathbf{x}}^{\mathbf{j}}$ is done either (1) by the additive construction (top block) where a shuffled version of the residuals is added to the predicted version using the remaining predictors with the mean of a random forest (RF) or (2) by the sampling construction (bottom block) using a random forest (RF) model to fit $\mathbf{x^j}$ from $\mathbf{X^{-j}}$ and then sample the prediction within the leaves of the RF.

## C  Conditional Permutation Importance (CPI) Wald statistic asymptotically controls type-I errors: hypotheses, theorem and proof

**Outline**    The proof relies on the observation that the importance score defined in (4) is $0$ in the asymptotic regime, where the permutation procedure becomes a sampling step, under the assumption that variable $j$ is not conditionally associated with $y$. Then all the proof focuses on the convergence of the finite-sample estimator to the population one. To study this, we use the framework developed in [Williamson et al., 2021]. Note that the major difference with respect to other contributions [Watson and Wright, 2021] is that the ensuing inference is no longer conditioned on the estimated learner $\hat{\mu}$. Next, we first restate the precise technical conditions under which the different importance scores considered are asymptotically valid, i.e. lead to a Wald-type statistic that behaves as a standard normal under the null hypothesis.

**Notations**    Let $\mathcal{F}$ represent the class of functions from which a learner $\mu : \mathbf{x} \mapsto y$ is sought.

Let $P_0$ be the data-generating distribution and $P_n$ is the empirical data distribution observed after drawing $n$ samples (noted $n_{train}$ in the main text; in this section, we denote it $n$ to simplify notations). The separation between train and test samples is actually only relevant to alleviate

some technical conditions on the class of learners used. $\mathcal{M}$ is the general class of distributions from which $P_1, \ldots, P_n, P_0$ are drawn. $\mathcal{R} := \{c(P_1 - P_2) : c \in [0, \infty), P_1, P_2 \in \mathcal{M}\}$ is the space of finite signed measures generated by $\mathcal{M}$. Let $l$ be the loss function used to obtain $\mu$. Given $f \in \mathcal{F}$, $l(f; P_0) = \int l(f(\mathbf{x}), y) P_0(\mathbf{z}) d\mathbf{z}$, where $\mathbf{z} = (\mathbf{x}, y)$. Let $\mu_0$ denote a population solution to the estimation problem $\mu_0 \in \operatorname{argmin}_{f \in \mathcal{F}} l(f; P_0)$ and $\hat{\mu}_n$ a finite sample estimate $\hat{\mu}_n \in \operatorname{argmin}_{f \in \mathcal{F}} l(f; P_n) = \frac{1}{n} \sum_{(\mathbf{x}, y) \in P_n} l(f(\mathbf{x}), y)$.

Let us denote by $\dot{l}(\mu, P_0; h)$ the Gâteaux derivative of $P \mapsto l(\mu, P)$ at $P_0$ in the direction $h \in \mathcal{R}$, and define the random function $g_n : \mathbf{z} \mapsto \dot{l}(\hat{\mu}_n, P_0; \delta_{\mathbf{z}} - P_0) - \dot{l}(\mu_0, P_0; \delta_{\mathbf{z}} - P_0)$, where $\delta_{\mathbf{z}}$ is the degenerate distribution on $\mathbf{z} = (\mathbf{x}, y)$.

**Hypotheses**

(A1) (Optimality) there exists some constant $C > 0$, such that for each sequence $\mu_1, \mu_2, \cdots \in \mathcal{F}$ given that $\|\mu_n - \mu_0\| \to 0, |l(\mu_n, P_0) - l(\mu_0, P_0)| < C\|\mu_n - \mu_0\|_{\mathcal{F}}^2$ for each $n$ large enough.

(A2) (Differentiability) there exists some constant $\kappa > 0$ such that for each sequence $\epsilon_1, \epsilon_2, \cdots \in \mathbb{R}$ and $h_1, h_2, \cdots \in \mathcal{R}$ satisfying $\epsilon_n \to 0$ and $\|h_n - h_\infty\| \to 0$, it holds that

$$\sup_{\mu \in \mathcal{F}: \|\mu - \mu_0\|_{\mathcal{F}} < \kappa} \left| \frac{l(\mu, P_0 + \epsilon_n h_n) - l(\mu, P_0)}{\epsilon_n} - \dot{l}(\mu, P_0; h_n) \right| \to 0.$$

(A3) (Continuity of optimization) $\|\mu_{P_0 + \epsilon h} - \mu_0\|_{\mathcal{F}} = O(\epsilon)$ for each $h \in \mathcal{R}$.

(A4) (Continuity of derivative) $\mu \mapsto \dot{l}(\mu, P_0; h)$ is continuous at $\mu_0$ relative to $\|.\|_{\mathcal{F}}$ for each $h \in \mathcal{R}$.

(B1) (Minimum rate of convergence) $\|\hat{\mu}_n - \mu_0\|_{\mathcal{F}} = o_P(n^{-1/4})$.

(B2) (Weak consistency) $\int g_n(\mathbf{z})^2 dP_0(\mathbf{z}) = o_P(1)$.

(B3) (Limited complexity) there exists some $P_0$-Donsker class $\mathcal{G}_0$ such that $P_0(g_n \in \mathcal{G}_0) \to 1$.

**Proposition** (Theorem 1 in [Williamson et al., 2021]) If the above conditions hold, $l(\hat{\mu}_n, P_n)$ is an asymptotically linear estimator of $l(\mu_0, P_0)$ and $l(\hat{\mu}_n, P_n)$ is non-parametric efficient.

Let $P_0^\star$ be the distribution obtained by sampling the j-th coordinate of $\mathbf{x}$ from the conditional distribution of $q_0(x^j | \mathbf{x}^{-\mathbf{j}})$, obtained after marginalizing over $y$:

$$q_0(x^j | \mathbf{x}^{-\mathbf{j}}) = \frac{\int P_0(\mathbf{x}, y) dy}{\int P_0(\mathbf{x}, y) dx^j dy}$$

$P_0^\star(\mathbf{x}, y) = q_0(x^j | \mathbf{x}^{-\mathbf{j}}) \int P_0(\mathbf{x}, y) dx^j$. Similarly, let $P_n^\star$ denote its finite-sample counterpart. It turns out from the definition of $\hat{m}_{CPI}^j$ in Eq. 4 that $\hat{m}_{CPI}^j = l(\hat{\mu}_n, P_n^\star) - l(\hat{\mu}_n, P_n)$. It is thus the final-sample estimator of the population quantity $m_{CPI}^j = l(\hat{\mu}_0, P_0^\star) - l(\hat{\mu}_0, P_0)$.

Given that $\hat{m}_{CPI}^j = l(\hat{\mu}_n, P_n^\star) - l(\hat{\mu}_0, P_0^\star) - (l(\hat{\mu}_n, P_n) - l(\hat{\mu}_0, P_0)) + l(\hat{\mu}_0, P_0^\star) - l(\hat{\mu}_0, P_0)$, the estimator $\hat{m}_{CPI}^j$ is asymptotically linear and non-parametric efficient.

The crucial observation is that under the j-null hypothesis, $y$ is independent of $x^j$ given $\mathbf{x}^{-\mathbf{j}}$. Indeed, in that case $P_0(\mathbf{x}, y) = q_0(x^j | \mathbf{x}^{-\mathbf{j}}) P_0(y | \mathbf{x}^{-\mathbf{j}}) P_0(\mathbf{x}^{-\mathbf{j}})$ and $P_0(x^j | \mathbf{x}^{-\mathbf{j}}, y) = P_0(x^j | \mathbf{x}^{-\mathbf{j}})$, so that $P_0^\star = P_0$. Hence, mean/variance of $\hat{m}_{CPI}^j$'s distribution provide valid confidence intervals for $m_{CPI}^j$ and $mean(\hat{m}_{CPI}^j) \underset{n \to \infty}{\to} 0$. Thus, the Wald statistic $\hat{z}_{CPJ}^j$ defined in section (4.2) converges to a standard normal distribution, implying that the ensuing test is valid.

In practice, hypothesis (B3), which is likely violated, is avoided by the use of cross-fitting as discussed in [Williamson et al., 2021]: as stated in the main text, variable importance is evaluated on a set of samples not used for training. An interesting impact of the cross-fitting approach is that it reduces the hypotheses to (A1) and (A2), plus the following two:

(B'1) (Minimum rate of convergence) $\|\hat{\mu}_n - \mu_0\|_{\mathcal{F}} = o_P(n^{-1/4})$ on each fold of the sample splitting scheme.

(B2') (Weak consistency) $\int g_n(\mathbf{z})^2 dP_0(\mathbf{z}) = o_P(1)$ on each fold of the sample splitting scheme.

# D   Computational scaling of *CPI-DNN* and leanness

Figure E2: *CPI-DNN* vs *LOCO-DNN*: Performance at detecting important variables on simulated data with $n = 1000$, $p = 50$ and $\rho = 0.8$ in terms of (**AUC score**), **Type-I error**, **Power** and **Time**. Dashed line: targeted type-I error rate. Solid line: chance level.

*Computationally lean* refers to two facts: (1) there is no need to refit the costly MLP learner to predict y unlike *LOCO-DNN* (A removal-based method provided with our learner) as seen in Fig. E2. Both *CPI-DNN* and *LOCO-DNN* achieved a high AUC score and controlled the Type-I error in a highly correlated setting ($\rho$=0.8). However, in terms of computation time, *CPI-DNN* is far ahead of *LOCO-DNN*, which validates our use of the permutation scheme. (2) The conditional estimation step involved for the conditional permutation procedure is done with an efficient RF estimator, leading to small time difference wrt *Permfit-DNN*; Overall we obtain the accuracy of LOCO-type procedures for the cost of a basic permutation scheme.

# E   Evaluation Metrics

**AUC score**   [Bradley, 1997]: The variables are ordered by increasing p-values, yielding a family of $p$ splits into relevant and non-relevant at various thresholds. AUC score measures the consistency of this ranking with the ground truth ($p_{signals}$ predictive features versus $p - p_{signals}$).

**Type-I error**   : Some methods output p-values for each of the variables, that measure the evidence against each variable being a null variable. This score checks whether the rate of low p-values of null variables exceeds the nominal false positive rate (set to 0.05).

**Power**   : This score reports the average proportion of informative variables detected (when considering variables with p-value $< 0.05$).

**Computation time**   : The average computation time per core on 100 cores.

**Prediction Scores**   : As some methods share the same core to perform inference and with the data divided into a train/test scheme, we evaluate the predictive power for the different cores on the test set.

# F  Supplement Figure 1 - Power & Computation time

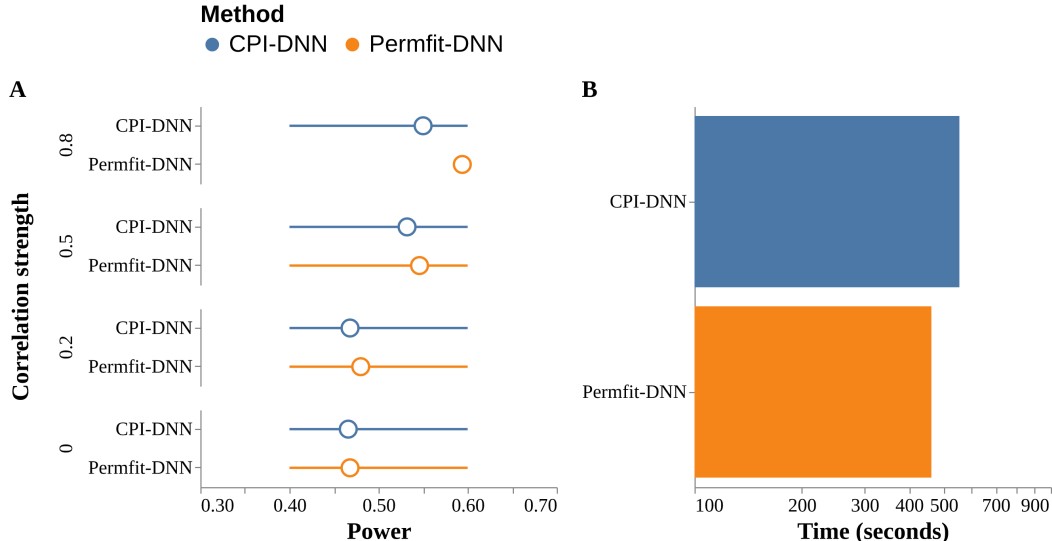

Figure 1 - S1: **Permfit-DNN vs CPI-DNN**: Performance at detecting important variables on simulated data under the setting of experiment 1, with $n = 300$ and $p = 100$. **(A)**: The power reports the average proportion of informative variables detected (p-value $< 0.05$). **(B)**: The computation time is in seconds with (log10 scale) per core on 100 cores.

Based on Fig. 1 - S1, both methods *Permfit-DNN* and *CPI-DNN* have almost similar power. In high correlation regime, Permfit-DNN yields more detections, but it does not control type-I errors (Fig. 1). Regarding computation time, *CPI-DNN* is slightly more computationally expensive than *Permfit-DNN*.

# G  Supplement Experiment 5.2 - Models

**Classification**  The signal $\mathbf{X}\boldsymbol{\beta}^{main}$ is turned to binomial variables using the probit function $\Phi$. $\boldsymbol{\beta}^{main}$ and $\boldsymbol{\beta}^{quad}$ are the two vectors with different lengths of regression coefficients having only $n_{\text{signal}} = 20$ non-zero coefficients, the true model. $\boldsymbol{\beta}^{main}$ is used with the main effects while $\boldsymbol{\beta}^{quad}$ is involved with the interaction effects. Following [Janitza et al., 2018], the $\boldsymbol{\beta}$ values $\in \{\boldsymbol{\beta}^{main}, \boldsymbol{\beta}^{quad}\}$ are drawn i.i.d. from the set $\mathcal{B} = \{\pm 3, \pm 2, \pm 1, \pm 0.5\}$.

$$y_i \sim Binomial(\Phi(\boldsymbol{x_i}\boldsymbol{\beta}^{main})), \ \forall i \in [\![n]\!]$$

**Plain linear model**  We rely on a linear model, where $\boldsymbol{\beta}^{main}$ is drawn as previously and $\epsilon$ is the Gaussian additive noise $\sim \mathcal{N}(0, \mathbf{I})$ with magnitude $\sigma = \frac{||\mathbf{X}\boldsymbol{\beta}^{main}||_2}{SNR\sqrt{n}}$: $y_i = \mathbf{x_i}\boldsymbol{\beta}^{main} + \sigma\epsilon_i, \ \forall i \in [\![n]\!]$.

**Regression with ReLu**  An extra ReLu function is applied to the output of the Plain linear model: $y_i = Relu(\mathbf{x_i}\boldsymbol{\beta}^{main} + \sigma\epsilon_i), \ \forall i \in [\![n]\!]$.

**Interactions only model**  We compute the product of each pair of variables. The corresponding values are used as inputs to a linear model: $y_i = \text{quad}(\mathbf{x_i}, \boldsymbol{\beta}^{quad}) + \sigma\epsilon_i, \ \forall i \in [\![n]\!]$, where $\text{quad}(\boldsymbol{x_i}, \boldsymbol{\beta}^{quad}) = \sum_{\substack{k,j=1 \\ k<j}}^{p_{signals}} \boldsymbol{\beta}^{quad}_{k,j} x_i^k x_i^j$. The magnitude $\sigma$ of the noise is set to $\frac{||\text{quad}(\boldsymbol{X},\boldsymbol{\beta}^{quad})||_2}{SNR\sqrt{n}}$. The non-zero $\boldsymbol{\beta}^{quad}$ coefficients are drawn uniformly from $\mathcal{B}$.

**Main effects with Interactions**    We combine both Main and Interaction effects. The magnitude $\sigma$ of the noise is set to $\frac{||\mathbf{X}\boldsymbol{\beta}^{\text{main}}+\text{quad}(\boldsymbol{X},\boldsymbol{\beta}^{quad})||_2}{SNR\sqrt{n}}$: $y_i = \mathbf{x_i}\boldsymbol{\beta}^{main} + \text{quad}(\mathbf{x_i}, \beta^{\mathbf{quad}}) + \sigma\epsilon_i, \ \forall i \in [\![n]\!]$.

# H    Supplement Figure 3 - Extended model comparisons

We also benchmarked the following methods deprived of statistical guarantees:

- Knockoffs [Candes et al., 2017, Nguyen et al., 2020]: The knockoff filter is a variable selection method for multivariate models that controls the False Discovery Rate. The first step of this procedure involves sampling extra null variables that have a correlation structure similar to that of the original variables. A statistic is then calculated to measure the strength of the original variables versus their knockoff counterpart. We call this the knockoff statistic $\mathbf{w} = \{w_j\}_{j=1}^{p}$ that is the difference between the importance of a given feature and the importance of its knockoff.

- Approximate Shapley values [Burzykowski, 2020]: SHAP being an instance method, we relied on an aggregation (averaging) of the per-sample Shapley values.

- Shapley Additive Global importancE (SAGE) [Covert et al., 2020]: Whereas SHAP focuses on the *local interpretation* by aiming to explain a model's individual predictions, SAGE is an extension to SHAP assessing the role of each feature in a *global interpretability* manner. The SAGE values are derived by applying the Shapley value to a function that represents the predictive power contained in subsets of features.

- Mean Decrease of Impurity [Louppe et al., 2013]: The importance scores are related to the impact that each feature has on the impurity function in each of the nodes.

- BART [Chipman et al., 2010]: BART is an ensemble of additive regression trees. The trees are built iteratively using a back-fitting algorithm such as MCMC (Markov Chain Monte Carlo). By keeping track of covariate inclusion frequencies, BART can identify which components are more important for explaining $\mathbf{y}$.

Based on AUC, we observe SHAP, SAGE and Mean Decrease of Impurity (MDI) perform poorly. These approaches are vulnerable to correlation. Next, Knockoff-Deep and Knockoff-Lasso perform well when the model does not include interaction effects. BART and Knockoff-Bart show fair performance overall.

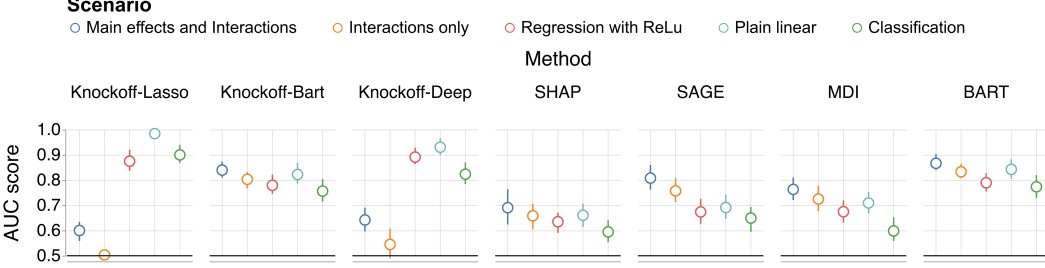

Figure 3 - S1: **Extended model comparisons**: State-of-the-art methods for variable importance not providing statistical guarantees in terms of p-values are compared (outer columns) and to competing approaches across data-generating scenarios (inner columns) using the settings of experiments 2 and 3. Prediction tasks were simulated with $n = 1000$ and $p = 50$. Solid line: chance level.

# I  Supplement Figure 3 - Power

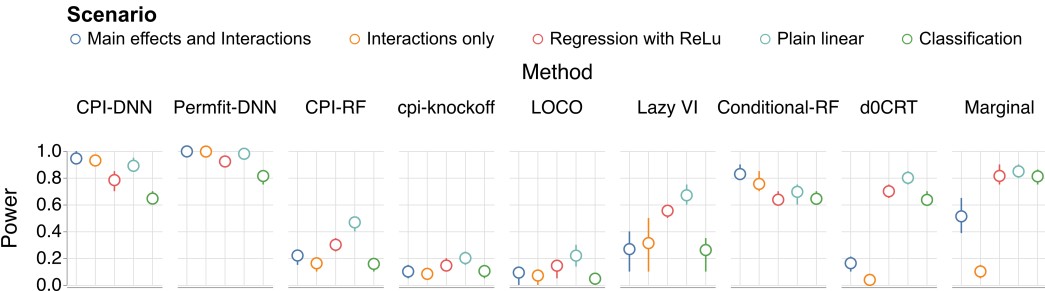

**Figure 3 - S2: Extended model comparisons**: *CPI-DNN* and *Permfit-DNN* were compared to baseline models (outer columns) and to competing approaches across data-generating scenarios (inner columns). Convention about power as in Fig. 1 - S1. Prediction tasks were simulated with $n = 1000$ and $p = 50$.

Based on the power computation, *Permfit-DNN* and *CPI-DNN* outperform the alternative methods. Thus, the use of the right learner leads to better interpretations.

# J  Supplement Figure 3 - Computation time

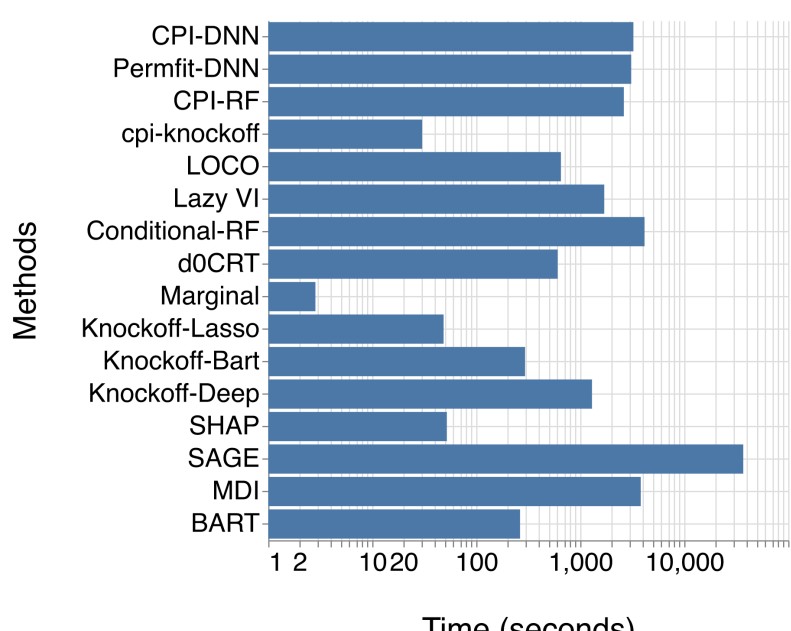

**Figure 3 - S3: Extended model comparisons**: The computation times for the different methods (with and without statistical guarantees in terms of p-values) are reported in seconds with (log10 scale) per core on 100 cores. Prediction tasks were simulated with $n = 1000$ and $p = 50$.

The computation time of the different methods mentioned in this work (with and without statistical guarantees) is presented in Fig. 3 - S3 in seconds with (log10 scale). First, we compare *CPI-RF*, *cpi-knockoff* and *LOCO* based on a Random Forest learner with $p=50$. We see that *cpi-knockoff* and *LOCO* are faster than *CPI-DNN*. A possible reason is that *CPI-DNN* uses an inner 2-fold internal validation for hyperparameter tuning (learning rate, L1 and L2 regularization) unlike the alternatives. Next, The DNN-based methods (*CPI-DNN* and *Permfit-DNN*) are competitive with the alternatives that control type-I error ($d_0CRT$, *cpi-knockoff* and *LOCO*) despite the use of computationally lean learners in the latter.

## K  Supplement Figure 3 - Prediction scores on simulated data

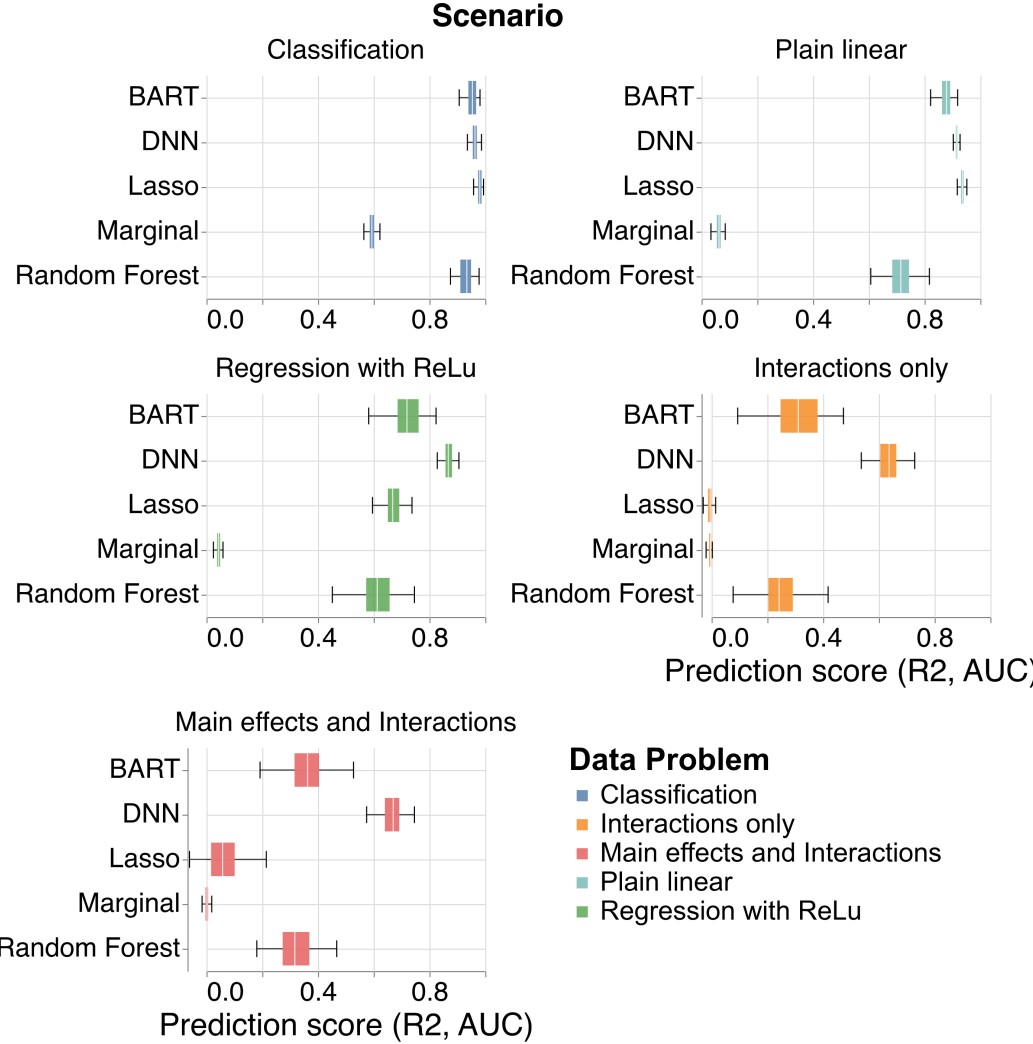

Figure 3 - S4: **Evaluating predictive power**: Performance of the different base learners used in the variable importance methods (**Marginal** = {Marginal effects}, **Lasso** = {Knockoff-Lasso}, **Random Forest** = {MDI, d0CRT, CPI-RF, Conditional-RF, cpi-knockoff, LOCO}, **BART** = {Knockoff-BART, BART} and **DNN** = {Knockoff-Deep, Permfit-DNN, CPI-DNN, Lazy VI}) on simulated data with $n$ = 1000 and $p$ = 50 in terms of **ROC-AUC** score for the classification and **R2** score for the regression.

The results for computing the prediction accuracy using the underlying learners of the different methods are reported in Fig. 3 - S4. Marginal inference, performs poorly, as it is not a predictive approach. Linear models based on Lasso show a good performance in the no-interaction effect scenario. Non-linear models based on Random Forest and BART improve on the lasso-based models. Nevertheless, they fail to achieve a good performance in scenarios with interaction effects. The models equipped with a deep learner outperform the other methods.

# L    Large scale simulations

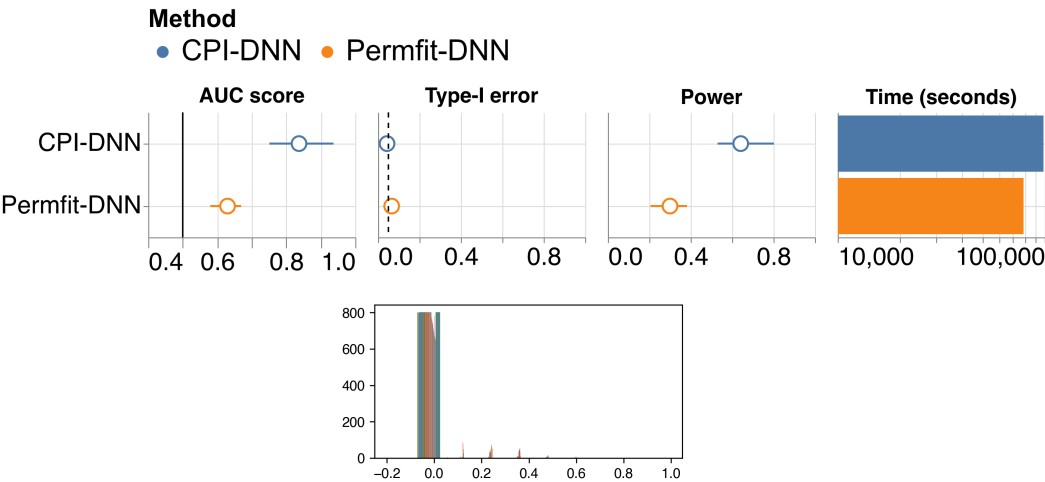

Figure E3: **Semi-simulation with UK Biobank**: **(Top panel)** Performance of *CPI-DNN* and *Permfit-DNN* is compared in terms of **AUC score**, **Type-I error**, **Power** and **Time** on the data from UKBB with $n = 8357$ and $p = 671$. **(Bottom panel)** Correlation strength among the variables in the UKBB dataset.

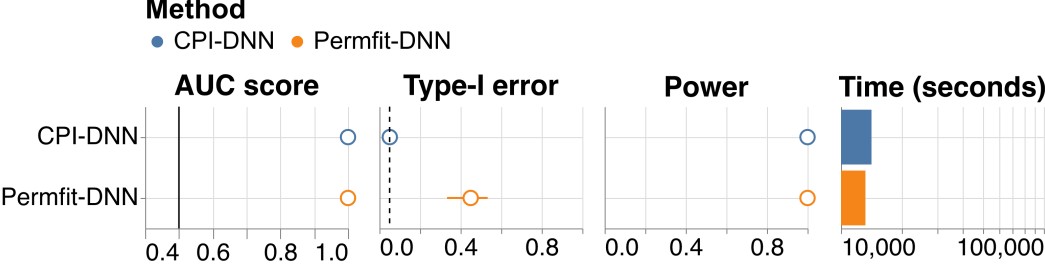

Figure E4: **Large scale simulation**: Performance of *CPI-DNN* and *Permfit-DNN* is compared in terms of **AUC score**, **Type-I error**, **Power** and **Time** on simulated data with $n = 10000$, $p = 50$ and $\rho = 0.8$.

In Figs. E3 and E4, we provide a comparison of the performance of both *Permfit-DNN* and *CPI-DNN* on the semi-simulated data from UK Biobank, with the design matrix consisting of the variables in the UK BioBank and the outcome is generated following a random selection of the true support, where $n$=8357 and $p$=671, and a large scale simulation with $n$=10000, $p = 50$ and block-based correlation of coefficient $\rho = 0.8$. For the UKBB-based simulation, we see that *CPI-DNN* achieves a higher AUC score and Power. However, both methods control the type-I error at the targeted level. To better understand the reason, we plotted (Fig. E3 Bottom panel) the histogram of the correlation values within the UKBB data: in this case, we consider a low-correlation setting which explains the good control for *Permfit-DNN*. In the large scale simulation where the correlation coefficient is set to 0.8, the difference is clear and only *CPI-DNN* controls the type-I error.

## M  Age prediction from brain activity (MEG) in Cam-CAN dataset

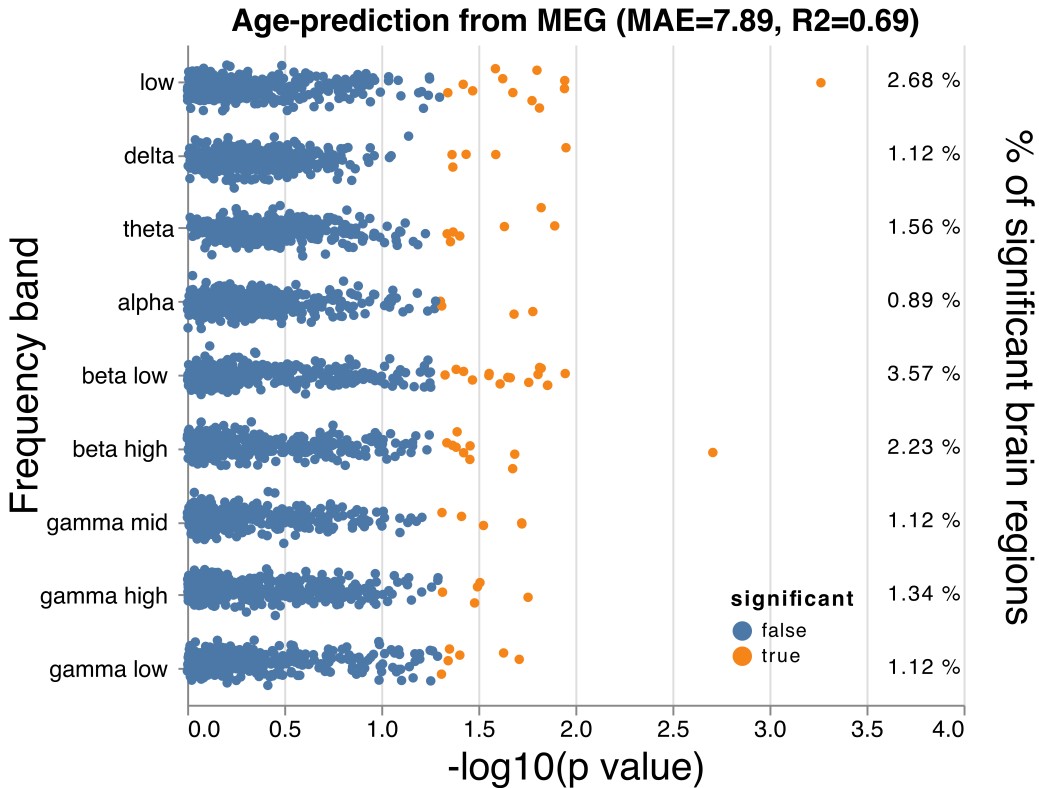

Figure E5: **Age prediction from brain activity**: Predicting age from brain activity in different frequencies with $n = 536$ and $p = 4032$.

Following the work of Engemann et al. [2020], we have applied *CPI-DNN* to the problem of age prediction from brain activity in different frequencies recorded with magnetoencephalography (MEG) in the Cam-CAN dataset. Without tweaking, the DNN learner reached a prediction performance on par with the published results as seen in Fig. E5. The p-values formally confirm aspects of the exploratory analysis in the original publication (importance of beta band).

# N Practical validation of the normal distribution assumption

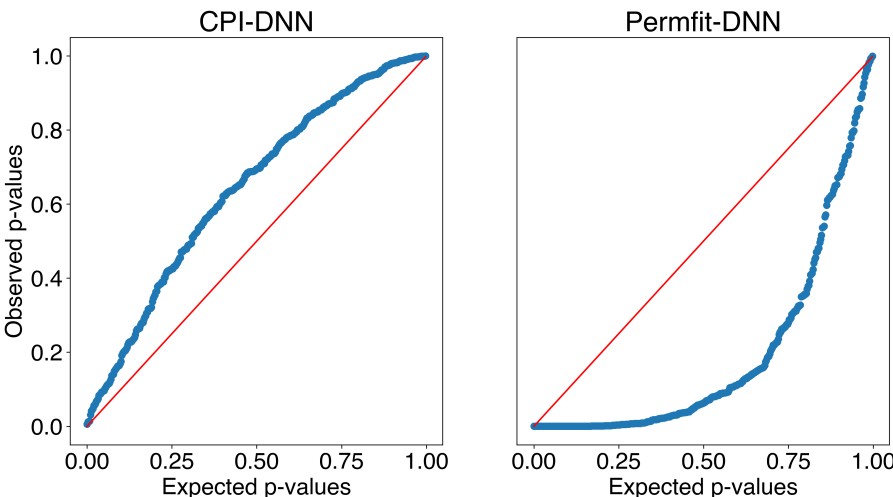

Figure E6: ***CPI-DNN* vs *Permfit-DNN* p-values calibration**: Q-Q plot for the distribution of the p-values vs the uniform distribution with $n = 1000$ and $p = 50$.

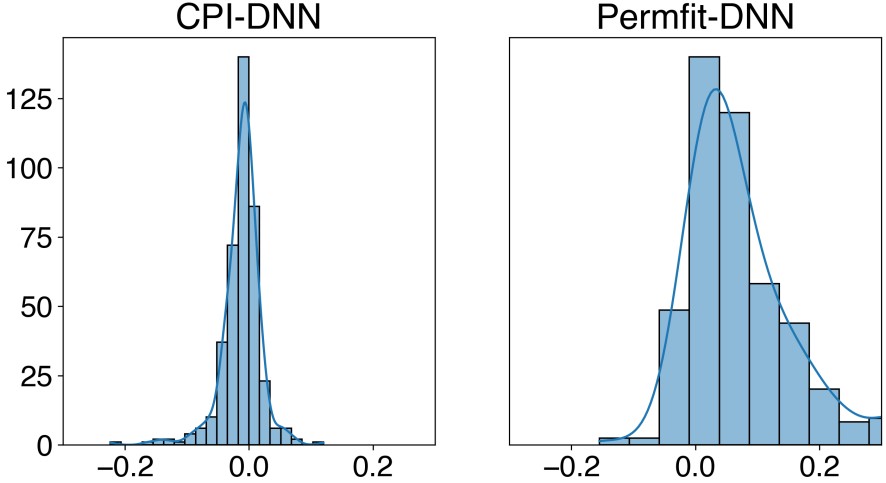

Figure E7: **Normal distribution assumption**: Histogram plots of the distribution of the importance scores of a random picked non-significant variable with $n = 1000$ and $p = 50$.

In Fig. E7, we compared the distribution of the importance scores of a random picked non-significant variable using *CPI-DNN* and *Permfit-DNN* through histogram plots, and we can emphasize that the normal distribution assumption holds in practice.

Also, in Fig. E6, we plot the distribution of the p-values provided by *CPI-DNN* and *Permfit-DNN* vs the uniform distribution through QQ-plot. We can see that the p-values for *CPI-DNN* are well calibrated and slightly deviated towards higher values. However, with *Permfit-DNN* the p-values are not calibrated.

## O    Random Forest for modeling the conditional distribution and resulting calibration

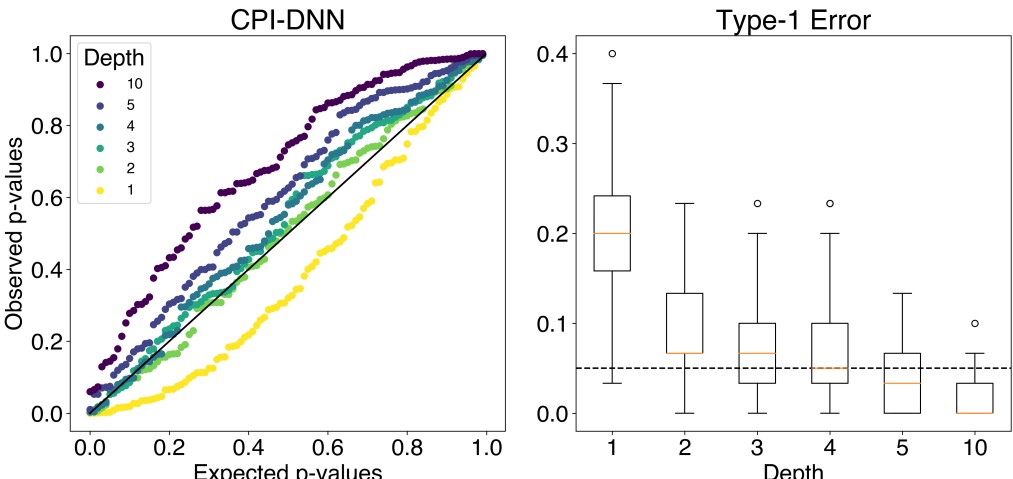

Figure E8: **Random forest calibration**: Calibration of the p-values for *CPI-DNN* (left panel) and the control of type-I error (right panel) as a function of the complexity of the Random Forest (the max depth of the trees). Dashed line: targeted type-I error rate. Solid line: uniform distribution.

The use of the Random Forest model was to maintain a good non-linear model with time benefits for the prediction of the conditional distribution of the variable of interest. In Fig. E8, We can see that reducing the depth to 1 or 2, thus making the model overly simple, breaks the control of the type-I errors at the targeted level. With larger depths, the model becomes more conservative. Therefore, the max depth of the Random Forest is chosen based on the performance with 2-fold cross validation.

