# OpenReview forum: "Statistically Valid Variable Importance Assessment through Conditional Permutations"
_NeurIPS.cc/2023/Conference — NeurIPS 2023 poster_

### Official Review · Reviewer_HX9z · 2023-06-21

**Soundness:** 3 good
**Presentation:** 4 excellent
**Contribution:** 3 good
**Rating:** 6
**Confidence:** 4

**Summary:**

In this work, the authors proposed a novel conditional permutation importance approach (CPI) that can well estimate the variable importance with the p-value provided under the condition that correlation exists across features. The authors theoretically justify the inflation of the feature permutation-based approach and justify the calibration of CPI. Empirically, CPI yields calibrated results across different settings with good power, which outperforms existing methods in the related field.

**Strengths:**

This paper focuses on addressing two key issues in interpretable machine learning: statistical calibration and scalability. The proposed method, CPI, shows a significant improvement in statistical calibration compared to the naive feature permutation-based approach. The paper also provides theoretical justification for CPI's ability to effectively control type-I errors. In simulation datasets, CPI outperformed other methods that addressed feature dependencies in terms of the AUC that measures the consistency of feature ranking.

**Weaknesses:**

The existing evidence is insufficient to convince me that CPI is well-calibrated. Several items can improve the soundness of the paper:
1. The authors should also provide the qqplot with a reasonable amount of tests (>=100) to demonstrate the calibration.
2. The authors used RF to approximate the feature j's conditional distribution; more justification is needed for why this is a good choice. (i.e., Does a simpler model achieve the same FPR and AUC level?)
3. The sample sizes in simulations are fairly small. It is unclear whether the well-controlled FPR was due to the calibration or the lack of power to detect inflation. A sample size of at least 10K is suggested w.r.t the given feature size: 50. If computation is a problem, I would suggest either precomputing the SVD of the covariance matrix and then transposing the data matrix to get $X$ with desired correlation. Or it would be even better if the authors could (also) perform the simulation on the UK Biobank genotype dataset (they have more realistic feature dependencies needed for this work).


**Questions:**

1. It is surprising to see the significant difference in AUC between CPI and cpi-knockoff. Do the authors have insight into what makes the difference?

2. Getting from equation (2) to equation (3) is not intuitive to me. It would be great if the authors could provide more detailed proof.

3. How did the authors compute the actual feature ranking? Which scheme did they use (i.e. Did you take the integration of equation (4); use Shapley value; or rank by coefficients?)



**Limitations:**

The authors have addressed several limitations of this work:
1. The limitation of using RF to approximate the conditional independence
2. The limitation of sample size for the trainer

Additionally, the theoretical calibration of the CPI relies on several hypotheses (A1, A2, A3, A4, B1, and B2) defined in the supplementary. It would be beneficial to have further discussion on the extent to which each hypothesis can be met in reality. Furthermore, if any of the hypotheses cannot be satisfied, it would be helpful to know the degree to which the calibration would be affected.

---

> ### Author Rebuttal · Authors · 2023-08-09
>
> W1: “The authors should also provide the qqplot with a reasonable amount of tests (>=100) to demonstrate the calibration.”
>
> R_W1: In $\textbf{PDF Fig. R1}$, we provided the qq-plot for the distribution of the p-values provided by Permfit-DNN and CPI-DNN for one randomly picked non-significant variable (the calibrated p-values should theoretically follow the uniform distribution). We can see that the p-values obtained by CPI-DNN are following the uniform distribution with a slight deviation towards higher p-values (which what we aim for as this variable is non-significant). However, with Permfit-DNN the p-values are not calibrated.
>
> W2: “The authors used RF to approximate the feature j's conditional distribution; more justification is needed for why this is a good choice. (i.e., Does a simpler model achieve the same FPR and AUC level?)”
>
> R_W2: Please refer to the global response (Random Forest for modeling the conditional distribution and resulting calibration - G2) for the discussion around the use of the Random Forest and its calibration.
>
> W3: “The sample sizes in simulations are fairly small. It is unclear whether the well-controlled FPR was due to the calibration or the lack of power to detect inflation. A sample size of at least 10K is suggested w.r.t the given feature size: 50. If computation is a problem, I would suggest either precomputing the SVD of the covariance matrix and then transposing the data matrix to get X with desired correlation. Or it would be even better if the authors could (also) perform the simulation on the UK Biobank genotype dataset (they have more realistic feature dependencies needed for this work).“
>
> R_W3: In $\textbf{PDF Fig. R4-a}$ and $\textbf{PDF Fig. R4-b}$, we provide a comparison of the performance of both Permfit-DNN and CPI-DNN on the semi-simulated data from UKBB where $n$=$8357$ and $p$=$671$, and a large scale simulation with $n$=$10000$, $p$ = $50$ and block-based correlation of coefficient $\rho$ = $0.8$. For the UKBB-based simulation, we see that CPI-DNN achieves a higher AUC score and Power. However, both methods control the type-I error at the targeted level. To better understand the reason, we plotted (just below) the histogram of the correlation values within the UKBB data and indeed we are faced with a low-level correlation setting which explains the good control for Permfit-DNN. In the large scale simulation where the correlation coefficient is set to 0.8, the difference is clear and only CPI-DNN controls the type-I error.
>
> Q1: “It is surprising to see the significant difference in AUC between CPI and cpi-knockoff. Do the authors have insight into what makes the difference?”
>
> R_Q1: We can refer to two facts: (1) cpi-knockff is based on the use of a random forest model with the ranger implementation while CPI-DNN is equipped with a DNN, which could explain its lack of power and (2) cpi-knockoff is based on the use of Knockoff sampling to replace the variable of interest. Thus, one possible reason for this difference is the quality of the knockoff sampling. As we know, people often generate Gaussian knockoffs but this could be inaccurate (as shown our process is more accurate). We have observed that even a relatively small distribution shift in knockoff generation can lead to large errors at inference time.
>
> Q2: “Getting from equation (2) to equation (3) is not intuitive to me. It would be great if the authors could provide more detailed proof.”
>
> R_Q2: We’ve added the following part in the supplement for further clarifications:
> $\mu = \mathbf{xw} = x^{j} w^{j} + \mathbf{x^{-j} w^{-j}} = \mathbf{x^{-j} w^{-j}}$ because $x^{j}$ is a null feature i.e. $w^{j} = 0$, and $\hat{\mu} = \mathbf{x\hat{w}}$ are linear functions therefore:
>
> $ \hat{m}^j
> = \frac{1}{n_{test}} \sum_{i=1}^{n_{test}}  (\hat{\mu}(\mathbf{x_i}) - \hat{\mu}(\mathbf{x_i^{(j)}})) (2\mu(\mathbf{x_i}) - \hat{\mu}(\mathbf{x_i}) - \hat{\mu}(\mathbf{x_i^{(j)}}) + 2 \varepsilon_i)$
>
> $= \frac{1}{n_{test}} \sum_{i=1}^{n_{test}}  (\cancel{\mathbf{x_{i}^{-j}} \mathbf{\hat{w}^{-j}}} + x_{i}^{j} \hat{w}^{j} - \cancel{\mathbf{x_{i}^{-j}} \mathbf{\hat{w}^{-j}}} - \\{x_{i}^{j}\\}^{perm} \hat{w}^{j}) (2 \mathbf{x_{i}^{-j}} \mathbf{w^{-j}} - 2 \mathbf{x_{i}^{-j}} \mathbf{\hat{w}^{-j}} - (x_{i}^{j}\hat{w}^{j} + \\{x_{i}^{j}\\}^{perm} \hat{w}^{j}) + 2 \varepsilon_i)$
>
> $= \frac{2\hat{w}^{j}}{n_{test}} \sum_{i=1}^{n_{test}}  (x_{i}^{j} - \\{x_{i}^{j}\\}^{perm}) (\mathbf{x_{i}^{-j}} (\mathbf{w^{-j}} - \mathbf{\hat{w}^{-j}})  + \varepsilon_i) - \cancel{\hat{w}^{j}((x_{i}^{j})^{2} - (\\{x_{i}^{j}\\}^{perm})^{2})}$
>
> $= \frac{2\hat{w}^{j}}{n_{test}} \sum_{i=1}^{n_{test}}  (x_{i}^{j} - \\{x_{i}^{j}\\}^{perm}) (\mathbf{x_{i}^{-j}} (\mathbf{w^{-j}} - \mathbf{\hat{w}^{-j}})  + \varepsilon_i)$
>
> Q3: “How did the authors compute the actual feature ranking? Which scheme did they use (i.e. Did you take the integration of equation (4); use Shapley value; or rank by coefficients?)”
>
> R_Q3: As described in Practical estimation (section 4.2), the equation 4 (in the regression case) is used through the loss score defined in equation 6 for B drawings. We thus compute a Wald-type statistic as stated L.179-181 by “dividing the mean of the importance scores by the corresponding standard deviation. P-values are computed using the cumulative distribution function of the standard normal distribution”. These p-values are used to perform the ranking which is equivalent to rank based on the Wald statistic or standardized importance scores.
>
> L1: “It would be beneficial to have further discussion on the extent to which each hypothesis can be met in reality. Furthermore, if any of the hypotheses cannot be satisfied, it would be helpful to know the degree to which the calibration would be affected.”
>
> R_L1: Please refer to the global response (Groups needed for very highly correlated settings - G3) where we provide the discussion related to the need for groups in very highly correlated settings.

---

> > ### Comment · Reviewer_HX9z · 2023-08-14
> >
> > The authors have detailedly addressed the majority of my concerns in their rebuttal.
> >
> > One interesting observation the authors have shown is that as the max depth of the tree increase, the FPR becomes gradually conservative. My takeaway is that model overfitting will lead to conservative results, while an underfitting model can lead to inflation. But it can be tricky, in practice, to find such an estimator that well-fits the data. It would be great to understand the reason behind this, and I suggest the authors further justify their algorithm's robustness in FPR control or acknowledge this limitation.
> >
> > However, I understand the limited time the authors have in addressing all my concerns. The revised article has shown improvement compared to the original version, therefore I will adjust my rate to 6.

---

> > > ### Author Response · Authors · 2023-08-17
> > >
> > > We thank the reviewer for the positive appraisal of the efforts we put in the revision. We agree that the subject of controlling the calibration of the p-values under the use of the Random Forest model as a conditional probability learner is important. We hope that the following consideration will help clarify how the depth experiment and our initial results are connected:
> > >
> > > In **PDF Fig.R3**, we demonstrated that an overly simplistic model (small depth) is losing its calibration and a more complex model (with high depth) is more conservative.
> > >
> > > Importantly, in **PDF Fig.R1** we reported satisfying calibration, which was not ad-hoc but achieved through hyper-parameter tuning of the Random Forest’s depth.
> > >
> > > Concretely, we found that, for our setting, 2-fold cross validation for the depth of the Random Forest model among a list of pre-defined depths {2, 5, 10} was satisfying to achieve this calibration.
> > >
> > > We used a small set of pre-defined depths not to lose the computational benefit of the method while maintaining the predictive accuracy. The 2-fold cross validation was performed for each sampling step (thus with new incoming data). To clarify the importance of tuning for model calibration, we will highlight the 2-fold cross validated depth for the Random Forest model and include these additional results in the supplementary material, so that the readers can benefit from the results of this conversation.

---

### Official Review · Reviewer_XsJL · 2023-07-05

**Soundness:** 3 good
**Presentation:** 3 good
**Contribution:** 2 fair
**Rating:** 4
**Confidence:** 4

**Summary:**

The authors argue for the importance of using conditional sampling in permutation importance (CPI) methods.
They show that when features are correlated, using non-conditional sampling for permutation importance (on a finite training set) leads to a bias in the estimated feature importance. (Specifically, the bias is discussed in the case of null features having positive importance).
As their main result, they show that under regularity conditions (similar to those of Williamson 2021), the conditional permutation importance estimator is approximately normal and therefore the resulting p-values / intervals would be valid.
They produce an implementation of CPI, which they test against the permfit-DNN algorithm of based on Mi et al in a series of experiments. Here the main result is that indeed the non-conditional permutations introduce bias (when interpreted as p-values) and a little loss in "sorting" accuracy (that is in the relative rank of the null examples).

**Strengths:**

Strengths:
Clarity: The paper is in general well written and quite easy to follow. The arguments and proofs are surveyed well. The experiments and results are easy to follow.

Quality:
The theoretical derivations are convincing and seem correct. The experimental results are well set to show the point that methods can break if they don’t account for the correlation of the variables. The comparison to other explainability methods is quite comprehensive in terms of the survey.


Novelty:
There are some novel aspects to this paper: first and foremost, the method implements a code base for producing conditional inference without assuming a specific type of correlation between the variables, and with the learning model implemented at the same time. This might make the method attractive to users.

In addition, we have two main theoretical results. The first is a detailed critical discussion of the assumptions of Mi et al, showing an unaccounted for bias term in their analysis. The analysis is nice, but I am not sure it holds the paper. The second is an extension of the Williamson result for condition permutation importance, showing that the resulting statistic is approximately normal under some conditions. IT is important for the estimator, but as far as I can tell, applies the proofs in the Williamson paper to our estimator.

However, here it should be said that in my view the paper significantly overclaims, and does not describe well the current knowledge, See more in “weaknesses”.



Significance: The problem of identifying important features that contribute to an outcome of interest is fundamental
to scientific research, in particular in the biosciences. Recent years have seen many methods that try to adapt
these ideas so they would work well for non-linear models. This paper proposes a new method (and software)
for computing conditional permutation importance, and provides some theoretical and empirical evidence for the superiority
of the conditional sampling over the marginal sampling. Because conditional sampling is harder, having out-of the box
code for performing this analysis may be important.

**Weaknesses:**

Weakness:

The main weakness of this paper in my view is that it is not honest enough in representing its contribution to the literature. At a first read, and perhaps for researchers who are not experts in the field, it may sound as if the authors were the first to propose conditional permutation importance. (See the structure of the introduction - the idea of conditional permutation importance is not mentioned, except under their contribution).
“””In this work, we introduce an improvement of permutation importance called conditional per46 mutation importance. This method uses a conditional sampling step to compute the conditional  importance, taking into account the correlation between the variables. In section 3 we analyze  the limitations of the permutation approach. In section 4 we describe a conditional sampling based variable importance framework and establish that it controls type-I errors. In section 5, we  systematically compare the most popular variable importance measures on various classification 51 and regression tasks, with a focus on type-I error control.”””
When they acknowledge other CPIs, in passing in Section 2, they only mention in the context of Strobl et al.

In practice, CPIs have long been discussed in the general framework of feature importance. See for example Chapter 8.2 of Fisher et al in JMLR. More generally, the debate whether features removed (in a simple removal as in Covert 2022, or as in SHAP) should be imputed from the marginal distribution or from the conditional distribution has been ongoing. For example, the original definition of SHAP is designed to sample conditionally, but for computational reasons resorts to marginal sampling. In Tree-SHAP, the sampling is conditional.

In implying that this paper introduces conditional permutation inference, the authors are also doing a disservice to the readers in neglecting to more clearly stake their actual innovations, allowing the reader to evaluate them. For example, is the Mi result important so that showing its fault is important? Articulate this.
(In my view, the most important part here is the software with out-of-box estimators for the conditional distribution.  But other arguments by the authors would make their case more clear, Currently, reading the abstract, I understand their main innovation to be introducing CPI, which obviously is not true.
(Comment: My current very low grade is due to this weakness. A satisfactory rewrite of this would probably allow me to increase the score to the area of just below or above acceptance).


Other than that, it seems to be a paper with rather modest innovation, outside of the software itself. Perhaps my view is dimmed by the overclaiming, and put in proper context I will be happy to re-evaluate.


3.Because in my view the most important part is the software tool, I would have loved to see more about how it works compared to other methods. In particular, more sophisticated relations between variables; how sensitive it is to an increase in p compared to other methods; An interesting question that goes unanswered is how much tweeking goes into making the method work well for new data; perhaps an analysis of multiple datasets would help here, (though I understand the added costs in writing that paper).


Other comments:

Out of the box methods for conditional inference could backfire if the relation between the variables is hard to model correctly, or if there are too many variables / not enough samples so that variables are almost completely determined by other variables. I would imagine that a conditional method would provide some quality control and safeguards regarding the accuracy of modeling one variable from the rest.

Both the type-1 error statistics and the AUC are defined in terms of telling appart important and unimportant variables. Is there any sense where we can understand the accuracy or variability in ranking for the non-null variables? Feature important values can usually be interpreted for non-null cases as well.

Smaller comments:

Figure 4: color scheme is not consistent between left and right figures (same colors, different meanings). On the right color mean intersection of model results. On the left, the same colors mean source of features, whereas the intersection of model results are marked by the symbols. Also, crosses and circles almost identical; top rank is 1 not p which should be mentioned; Right side is a bit hard to parse.

#### Changes ####

Following discussions, the authors suggested specific changes to the writing and to the citations that I believe better reflect their contribution to the field. I will therefore update the score to borderline reject.

**Questions:**

See above.

**Limitations:**

I think one limitation not discussed is the difficulty in modeling the conditional distribution, and what happens if there are shortcomings in this model.

(No societal impact problems :)

---

> ### Author Rebuttal · Authors · 2023-08-09
>
> S1: “The first is a detailed critical discussion of the assumptions of Mi et al (...) ”
>
> R_S1: Mi et al. 2021 demonstrated the intuitive estimation of variable importance in the biomedical context using an effective brute-force approach. This can work well thanks to their high-capacity function approximator (DNN), yet, lacks statistical guarantees (under the standard permutation approach). Our work shows how to overcome the statistical failure modes of Mi et al. while preserving their effective DNN model.
>
> S2: “ The second is an extension of the Williamson result (...).”
>
> R_S2: Williamson et al. 2021 provides the general framework for variable importance based on the removal of the variable of interest with a theoretical proof. We used their regularity conditions mentioned in L.152 and Appendix A. We carefully translated the proof to our setting (large-scale biomarker discovery). Importantly, we found that, in our case  (L.178-179) “unlike Williamson (...), the variance estimator is non-vanishing, and thus can be used as a plugin”, which adds support to the validity of our usage of this result in our setting.
>
> W1: “The main weakness of this paper in my view is that it is not honest enough (...).“
>
> R_W1: The reviewer flags a potential misunderstanding. We have studied previous contributions and did not intend to claim inventorship for the CPI idea. In Related Work (section 2) L.77-89, we discuss Strobl et al. 2008 (exclusive for Random Forests), Candes et al. 2017 (the variable of interest is sampled conditionally on the other covariates multiple times to compute a test statistic and p-values) and Watson and Wright 2021 (the necessity of conditional schemes via knockoff sampling).
> To address the concern, we carefully rephrased introduction, related work and method sections to clarify how we build on top of the CPI literature.
> In fact, we have not found any deep discussion of the practical implications of using CPI methods, nor a proof of the validity that would hold unconditionally with respect to estimated learners $\hat{\mu}$. Neither have we found systematic reusable benchmarks of CPI approaches against alternatives. An important suggestion made in our work is that a relatively efficient conditional sampling schema can be used, rendering CPI more affordable.
>
> W2: “ In implying that this paper introduces conditional permutation inference, (...) is the Mi result important (...)? (...).“
>
> R_W2: Our work studies the properties of CPI techniques, leading to a more general algorithm based on a thorough analysis of the limitations of the standard permutation scheme. To extend CPI to support the family of DNN learners, i.e. to make the proof more general, we adapted Williamson’s proof to our work as shown in Appendix A. Our work provides systematic benchmarks for CPI methods and helps clarify their respective advantages and failure modes. Of note, the LOCO approach in our work implements the method of Williamson et al 2021 and performed worse than the proposed CPI method, cf. Fig 3 in main text, $\textbf{PDF Figs R4-6}$ (rebuttal), which is non-trivial.
> We will provide a usable general implementation supporting any scikit-learn compatible estimators as base learners and conditional probability learners. The work of Mi et al. was inspiring, as it provides a state-of-the-art estimator with strong AUC-scores (cf. R_S1 above).
>
> W3: “Because in my view the most important part is the software tool, (...) multiple datasets would help here”
>
> R_W3: We thank the reviewer for the enthusiasm. Our work provides experts in variable importance and practitioners a reusable toolkit for composing and testing novel combinations of permutation algorithms and conditional probability estimators across various data-generating scenarios.
> To make our tools more usable, our GitHub repository will provide a high-level notebook upholding the usage of different methods, including CPI algorithms. Inspired by the reviewer, we have applied CPI-DNN to the problem of age prediction from brain activity in different frequencies recorded with magnetoencephalography (MEG) in the Cam-CAN dataset (cf. Engemann et al 2020, eLife). Without tweaking, the DNN learner reached a prediction performance on par with the published results ($\textbf{PDF Fig. R5}$). The p-values formally confirm aspects of the exploratory analysis in the original publication (importance of beta band). We have included this result in the paper.
>
> W4: “Out of the box methods for conditional inference (...) determined by other variables.”
>
> R_W4: We mention among the limitations (section 7) L.311 that the prediction could also “lead to inaccuracies and biased inference if the number of variables becomes too large”. To explore a related point, we have studied the model calibration when systematically reducing the random forest depth ($\textbf{PDF Fig. R3}$), which could provide a tool to study such issues in practice.
>
> W5: “Is there any sense where we can understand the accuracy or variability (...)”
>
> R_W5: The effect size corresponds to the effect on y that can be attributed to variable $x_{j}$, conditionally to the others. Since we have the variance, we can define a confidence interval associated with it. We have added a note to the main text to clarify this point.
>
> W6: “Figure 4: color scheme is not consistent between left and right figures (...)”
>
> R_W6: Indeed. To avoid confusion, we have now chosen two distinct color sets.
>
> L1: “I think one limitation not discussed is the difficulty in modeling the conditional distribution (...)”
>
> R_L1: We now mention this limitation in the main text. To explore it, we have studied model calibration when systematically reducing the random forest depth ($\textbf{PDF Fig. R3}$)-Global response G2, showing how a shallow random forest model actually breaks the model calibration while higher capacity increases conservatism. In practice, the depth is under control of grid search. We have added the result to the paper (supplement).

---

> > ### Comment · Reviewer_XsJL · 2023-08-17
> > **״The reviewer flags a potential misunderstanding. ״**
> >
> >
> > I thank the authors for their answer.
> >
> > As far as I can tell, the issue with moving between unconditional and conditional sampling of the missing features seems to be well discussed in the literature, and as I said before I found the presentation in this paper lacking.
> > One place to see this is in "Debeer, D., Strobl, C. Conditional permutation importance revisited. (2020)", which I think illuminates how the conditional and marginal importance reflect two different definitions as to the meaning of feature importance more broadly than in Random Forest.
> > Another example for this discussion (in the CS side of the literature) can be seen in the SHAP papers which claim that it is better to sample the imputed features from the conditional distributions (though sometimes they settle for the marginals). They are critiqued by Merrick and Taly (The Explanation Game: Explaining Machine Learning Models Using Shapley Values, 2020, page 6), who claim that conditional sampling may lead to artificially inflated importance. (I think Debeer and Strobl's perspective may help understand this discussion).
> > My point is that both marginal and conditional sampling for missing features are well known in the literature, and that their merits have been discussed. I think a new paper should acknowledge these discussions and be specific about the new contributions.
> >
> > ״״״ To address the concern, we carefully rephrased introduction, related work and method sections to clarify how we build on top of the CPI literature. ״״״
> >
> > As I found this issue to prevail in the abstract and throughout the paper (as detailed in my review), could you be specific regarding the rephrasing?
> >
> > Thank you !

---

> > > ### Author Response · Authors · 2023-08-18
> > > **Elaborating conceptual views and proposed changes to the main text**
> > >
> > > We understand that the reviewer wants to ensure that this potential misunderstanding be resolved and so do we. We would first like to respond to the **conceptual point** regarding marginal and conditional importance.
> > >
> > > * It is important that conditional inference is well defined mathematically, but our point of departure is that the definition of CPI lacks consensus. In our work we proposed a more general formulation alongside a flexible implementation, which has allowed us to perform a more systematic study of the properties of different permutation schemes and variable importance estimation methods that was lacking so far. Indeed, both conditional and unconditional inference schemes can be useful, which is consistent with some of our results for settings with low to moderate correlation.
> > > * That permutation-based approaches are widely used does not mean they are without problems (as developed in section 2, Related work). The work by Debeer et al. (2020), emphasize the limitations of permutation approaches as follows : “Simulation studies (Nicodemus et al. 2010) have indeed shown that even when there is no dependence between the outcome and any of the predictors ($X_k ⊥ Y$ holds for all $X_k$ ), highly correlated predictors (i.e., $X_k ⊥ Z_{(−k)}$ does not hold) have a positive PI (...) when the outcome is independent from the predictors, the CPI does not show the preference for correlated predictors that has been observed in the PI.” We thank the reviewer for having made us aware of this paper, which we have now included in our literature review.
> > >
> > > ## Changes:
> > > Our continuing conversation gives us the opportunity to highlight some of the, potentially, most impactful changes we propose for the revision (which was beyond the character limits of the initial rebuttal).
> > >
> > > ### In abstract:
> > > We change “Here we present a Conditional Permutation Importance approach (CPI) that is both model agnostic and computationally lean.” to
> > >
> > > “Here we develop a systematic approach for studying Conditional Permutation Importance (CPI) that is model agnostic and computationally lean alongside reusable benchmarks of state-of-the-art variable importance estimators.”
> > >
> > > ### In Introduction:
> > > We reorganized parts of the introduction to form a paragraph explicitly stating the contributions:
> > >
> > > * In this work, we propose a general methodology for studying the properties of Conditional Permutation Importance in biomedical applications alongside tools for benchmarking variable importance estimators:
> > >
> > >   * Building on the previous literature on CPI, we develop theoretical results for the limitations regarding Permutation Importance (PI) and advantages of conditional Permutation Importance (CPI) given correlated inputs.
> > >
> > >   * We propose a novel implementation for CPI allowing us to combine the potential advantages of highly expressive base learners for prediction (a deep neural network) and a comparably lean Random Forest model as a conditional probability learner.
> > >
> > >   * We conduct extensive benchmarks on synthetic and heterogeneous multimodal real-world biomedical data tapping into different correlation levels and data-generating scenarios for both classification and regression.
> > >
> > >   * We propose a reusable library for simulation experiments and real-world applications of our method on a public Github repo {link provided in the paper}“
> > >
> > > As a result, we have removed the potentially misleading sentence “In this work, we introduce an improvement of permutation importance called conditional permutation importance.“ from the paper to avoid the misunderstanding that we invented CPI.
> > >
> > > ### In sec 2 related work:
> > >
> > > As a result of our conversation, we have discovered additional relevant references, which we included in section 2 (Related work):
> > > * Hooker et al. (2021), Statistics and Computing;
> > > * Molnar et al. (2021), International Workshop on Extending Explainable AI Beyond Deep Models and Classifiers.
> > >
> > > We conclude the section: “In summary, previous work has established potential advantages of conditional permutation schemes for inference of variable importance. Yet, the lack of computationally scalable approaches within a unified framework has hampered systematic investigations of different permutation schemes in comparison to alternative techniques across a broader range of predictive modeling settings.”
> > >
> > > ### In section 4.2 (Practical estimation):
> > > We change “we describe how to compute conditional permutation importance”,  to
> > >
> > > “we present our proposed method for computing conditional permutation importance”
> > >
> > > ### In Discussion:
> > > We changed “In this work, we introduced *CPI-DNN* for estimating predictive variable importance, which was inspired by the limitations of the *Permfit-DNN* approach.” to
> > >
> > > “In this work, we developed a framework for studying the behavior of marginal and conditional permutation methods and proposed the *CPI-DNN* method, which was inspired by the limitations of the *Permfit-DNN* approach.” to avoid the misunderstanding that we invented CPI.

---

> > > > ### Comment · Reviewer_XsJL · 2023-08-19
> > > > **Changes proposed by the authors**
> > > >
> > > > Hi,
> > > > I thank the authors for their proposed changes.
> > > > I think they do a better job now in describing their contributions, and allow me to remove my strong insistence on rejecting
> > > > and increase the score to 4 (borderline reject).
> > > >
> > > > Two more side comments on this:
> > > > - In the reading of the original paper, it was not clear to me they were looking for a feature importance measure for biomedical
> > > > applications. I think I now better realize their  choice of data examples and motivations.
> > > > - In reading your quote from Debeer, it seems to make sense to cite Nicodemus et al. 2010 as well, as a paper that highlights the issue of using unconditional permutation.

---

> > > > > ### Author Response · Authors · 2023-08-20
> > > > > **Side comments**
> > > > >
> > > > > We thank the reviewer for his insightful comments and we are happy that the proposed changes helped to address the misunderstanding flagged in the initial response.
> > > > >
> > > > > For the side comments:
> > > > > * As for the applied context of the paper, we want to point out that in the abstract we mention “An empirical benchmark on real-world data analysis in a large-scale medical dataset showed that CPI provides a more parsimonious selection of statistically significant variables.”  and “Biomarker development is increasingly focusing on multimodal data including brain images, genetics, biological specimens and behavioral data“ (L.19-21) in the Introduction
> > > > > * Deber et al.’s work citing the work by Nicodemeus et al. 2010 is indeed a great work to add to our literature study. Yet, we’ve also mentioned two important works that highlights better the drawbacks of the use of non-conditional permutation, which we reported citing in our last response (we could not elaborate details given the character limitations):
> > > > >     * “While naı̈ve permutation-based methods can be appealing, we have shown with some quite simple examples that they can give misleading results. We also identify the extrapolation behavior by flexible models as a significant source of error in these diagnostics. The precise biases that permute-and-predict methods produce will depend on the learning method employed as well as the specifics of the dependence of the features and the response function; see Bénard et al (2021) for approaches to this analysis.“, the work by Hooker et al.(2021) “Unrestricted Permutation forces Extrapolation: Variable Importance Requires at least One More Model or There Is No Free Variable Importance“
> > > > >     * “When features are dependent, perturbation-based IML methods such as PFI, PDP, and Shapley values extrapolate in areas where the model was trained with little or no training data, which can cause misleading interpretations.“ section 5.1 in the work by Molnar et al. (2021) “General Pitfalls of Model-Agnostic Interpretation Methods for Machine Learning Models”

---

### Official Review · Reviewer_pFgF · 2023-07-07

**Soundness:** 3 good
**Presentation:** 3 good
**Contribution:** 3 good
**Rating:** 6
**Confidence:** 3

**Summary:**

The authors present an approach that aims to tackle the problem with the default permutation importance technique of performing variable importance. The default setting is prone to detecting correlated variables even as important even when they are not (Type I error). The authors present Conditional permutation importance: an approach that fixes this problem by conditionally sampling the variables based on the correlations. The results indicate the method is robust to type I errors, model agnostic and computationally efficient.

**Strengths:**

The paper has a number of strengths including a novel approach to solving the correlation issues with standard permutation importance methods, a rigorous analysis of the proposed method with clear results. The paper is very well written and formatted leading to a well oriented reading. The related works are discussed appropriately with similar methods compared as baselines. Its value to the field is clear and its applications broad. This is a strong paper.

**Weaknesses:**

- The clarity of the experiments can be improved. Firstly the selections of models, datasets and settings can be motivated better. Secondly, the first paragraph of section 5 mentiones that “default behavior” was implemented. It’s not very clear what this default behavior is.
- The contributions mention a computationally lean method, is this analysis presented somewhere? I cannot see it in the paper.

**Questions:**

- The figure text is a bit small and hard to read if not zoomed in on a PDF
- The paragraph formatting is strange is some places. For examples compare the strange linebreak in lines 26-27 with the standard Neurips format seen between paragraphs 74-75.
- Is permutation performance (general setting) prone to type II errors for certain models? One can imagine a hypothetical setting where a variable, age, is duplicated, i.e. full correlation between the two variables. A certain type of model can learn to predict solely based on “age copy”, since it includes all the required information and “age original” is redundant. “Age original” will therefore have 0 importance from a PI analysis. Are these not considered as the models that collapse onto one feature are rare?

**Limitations:**

There is a notable lack of discussion of the limitations of this method. One of the few weaknesses. The authors have many discussion points to touch on: computational complexity, sensitivity to the data structure, missing data, types of correlation, model configuration, just to name a few.

The discussion section is fairly short, with a very shallow discussion on the limitations. With this part expanded and made more comprehensive, this paper could go from strong to very strong.

---

> ### Author Rebuttal · Authors · 2023-08-09
>
> W1: “The clarity of the experiments can be improved. Firstly the selections of models, datasets and settings can be motivated better. Secondly, the first paragraph of section 5 mentioned that “default behavior” was implemented. It’s not very clear what this default behavior is.”
>
> R_W1: First of all, to improve clarity and readability, we have merged the sections 5-6 (Experiments and Results). In Related Work (section 2) L.90-92, we review the state of the art giving rise to the models selected for our work. As a result, we found strong interest in DNNs as function estimators in permutation algorithms. In Experiment 3 (section 5.3) L.222-223, we state that “we include Permfit-DNN and CPI-DNN in a benchmark with other state-of-the-art methods for variable importance.“ Our benchmarks include all major ideas presented in our review. We substantially extended the basic synthetic data setting from Mi et al. 2021 to include different degrees of correlation, non-linearities and prediction tasks. The UK Biobank example was chosen as a recent example (Dadi et al 2021) of predicting real-world outcomes from heterogenous, potentially correlated inputs. By “default behavior”, we wanted to highlight the fact that the different methods were used according to their original implementations without any modifications. As for Permfit-DNN, CPI-DNN and CPI-RF, the default implementation is based on a special internal 2 fold-cross validation for the significance study. As an improvement, we’ve changed “the default behavior of the different methods is implemented in order to maintain a fair comparison” to “we refer to the original implementation of the different methods in order to maintain a fair comparison”
>
> W2: “The contributions mention a computationally lean method, is this analysis presented somewhere? I cannot see it in the paper.”
>
> R_W2: Please refer to the global response (Regarding the computational scaling of CPI-DNN and leanness - G1) where we are introducing the new plots that explain the reason we used this term in our paper also (Random Forest for modeling the conditional distribution and resulting calibration - G2) for the calibration of the Random Forest model used to predict the conditional distribution of the variable of interest.
>
> Q1: “The figure text is a bit small and hard to read if not zoomed in on a PDF”
>
> R_Q1: We increased the font sizes in order to maintain a better experience for the reader.
>
> Q2: “The paragraph formatting is strange in some places. For example, compare the strange line break in lines 26-27 with the standard Neurips format seen between paragraphs 74-75.”
>
> R_Q2: The paragraph formatting is corrected following the NeurIPS standards.
>
> Q3: “Is permutation performance (general setting) prone to type II errors for certain models? One can imagine a hypothetical setting where a variable, age, is duplicated, i.e. full correlation between the two variables. A certain type of model can learn to predict solely based on “age copy”, since it includes all the required information and “age original” is redundant. “Age original” will therefore have 0 importance from a PI analysis. Are these not considered as the models that collapse onto one feature are rare?”
>
> R_Q3: Please refer to the global response (Groups needed for very highly correlated settings - G3) where we provide the discussion related to the need for groups in very highly correlated settings.
>
> L1: “There is a notable lack of discussion of the limitations of this method. One of the few weaknesses. The authors have many discussion points to touch on: computational complexity, sensitivity to the data structure, missing data, types of correlation, model configuration, just to name a few. The discussion section is fairly short, with a very shallow discussion on the limitations.”
>
> R_L1: We thank you for bringing this comment around the discussion part. We rephrase this section to include the following ideas: (1) In the global response, we conduct the response related to the computation scaling of CPI-DNN and why it is a “computationally lean” method - G1 therefore, we can add that “CPI-DNN has a linear and quadratic complexity as a function of the number of samples and variables respectively as seen in the Supplement”, while adding the $\textbf{PDF Fig. R6}$ to the Supplement. (2) “The efficiency of the CPI method is affected by the increase in the correlation coefficient between the variable i.e. two very high correlated variables will have a conditional importance of 0. Therefore, as a future direction, we suggest the use of group-based feature importance where groups of highly correlated variables are pre-defined”. (3) “Modeling the conditional distribution of the variable of interest is of great impact in this work where there is a need for a good non-linear model with time benefits. Random Forest is a good example which, given reasonable default choices implemented in standard software, benefits largely from tuning the depth of its trees as a single hyperparameter” - we’ve also added the $\textbf{PDF Fig.R3}$ to show the impact of the depth on the calibration of the results. (4) We do mention in the Discussion part (section 7) L.311 that the prediction could also “lead to inaccuracies and biased inference if the number of variables becomes too large”.

---

> > ### Comment · Reviewer_pFgF · 2023-08-20
> >
> > I thank the authors for taking my feedback into consideration. I will be keeping my overall rating.

---

### Official Review · Reviewer_2imb · 2023-07-08

**Soundness:** 3 good
**Presentation:** 3 good
**Contribution:** 3 good
**Rating:** 7
**Confidence:** 4

**Summary:**

The paper proposes a statistical testing framework for more effectively determining variable importance in generic learning settings. Using a conditional permutation test, they show that correlated variables are better handled theoretically and empirically compared to existing unconditional setups.

**Strengths:**

The proposed test is asymptotically and practically efficient, and while some of the theory (and assumptions) may seem strong, the resulting algorithm and test procedure are extremely clean. They take advantage of an interesting existing result in [Williamson et al 2021] and show that it can be implemented quite directly in modern learning settings.

The summary of related work seems complete.

The experimental evaluation is strong in supporting the claim that conditional importance is relevant and taking it properly into account leads to performance gains with respect to identifying variable importance compared to unconditional approaches, as well as the suite of selected other methods in the space.

**Weaknesses:**

1. It's unclear to me how the performance of the method would stack up against aggregrated Shapley or LIME-like methods. As the authors say they are quite popular, and while "local" or "instance-based", it would be interesting to see if some naive (or sophisticated) application of those is interesting here or not.

2. Figure 4 and the related discussion around L283 is somewhat difficult to parse. It's also not clear what is meant to be taken away from this; is one in better agreement with scientific consensus? What is the interpretation here?

3. The discussion of computation time in Section 7 suggests that it is significant. The result in the supplement suggests it was an additional hour or so (~450-525 min). How does this scale with number of variables? Number of samples? The abstract claims "computationally lean", I think if the authors want to state this, support is necessary in the main paper.

Minor:

4. The organization of the experimental section makes it quite difficult to read. I would suggest merging Sections 5 and 6 so that a reader does not need to flip back and forth between experimental setup and result.

5. Might be worth summarizing the power result from the Supplement in the main paper.

6. A few of the references are missing identifying information such as arXiv or DOI links, or venue.

**Questions:**

How good is the normal distribution assumption in practice? It might be interesting or valuable to plot the estimated null distribution in a few different settings with no variable importance.

I'm leaning towards acceptance for this work, if the above issues are well addressed. It is quite neat, and exposed this reviewer to a number of interesting ideas.

**Limitations:**

It's not clear if some version of existing Shapley/LIME/etc estimators may achieve similar results.

The efficiency claim is not well evaluated and does not seem true.

---

> ### Author Rebuttal · Authors · 2023-08-09
>
> W1: “​It's unclear to me how the performance of the method would stack up against aggregated Shapley or LIME-like methods. (...).”
>
> R_W1: In Results (section 6) L.278-280, we mention that “Additional benchmarks on popular methods that do not provide p-values, e.g. BART [Chipman et al., 2010] or Shapley values [Kumar et al., 2020], are reported in the supplement (section E)". In this plot, we examined the AUC score for an aggregated version of SHAP (Shapley values) also for SAGE - an extension for SHAP within a global interpretability manner - and we’ve observed that both methods perform rather poorly across the data-generating scenarios. We now highlighted in the text that we also benchmarked local and instance-based methods.
>
> W2: “Figure 4 and the related discussion around L283 is somewhat difficult to parse. It's also not clear what is meant to be taken away from this; is one in better agreement with scientific consensus? What is the interpretation here?”
>
> R_W2: We thank the reviewer for pointing out the opportunity for clarification. The analysis is motivated by the idea that it can be worthwhile revisiting an exploratory analysis of variable importance with two methods based on the same estimator, one safer method with guarantees and a “naive” approach, which may help detect debatable cases of spurious importance.
>
> We have modified the main text to explain the context better (first sentence under Experiment 4): “We conducted an empirical study of variable importance in a biomedical application using the non-conditional permutation approach Permfit-DNN (no statistical guarantees for correlated inputs) and the safer CPI-DNN approach. Differences between the methods might point at the confounding presence of correlations between inputs whereas consensus between the methods would point at safer conclusions of importance.”
>
> We’ve extended the results section to better highlight interesting cases of divergences between the methods: “As expected by the impact of aging on brain structure and function, brain data was most important for age-prediction compared to other outcomes. Interestingly, most disagreements between the methods occurred in this setting as CPI rejected 16 out of 66 brain inputs that were found as important by Permfit. This may point at the importance of correlations between brain variables, that may provoke spurious importance findings with Permfit.'.
>
> W3: The discussion of computation time in Section 7 suggests that it is significant. The result in the supplement suggests it was an additional hour or so (~450-525 min). How does this scale with the number of variables? Number of samples? The abstract claims "computationally lean", I think if the authors want to state this, support is necessary in the main paper.
>
> R_W3: This was due to a bug with the plotting (cf. global response - G4). We’ve corrected the time unit in the supplement figures, it should be in seconds rather than minutes. Therefore, we are talking here about a difference of up to 1 min. Please refer to the global response for the response regarding the computation scaling of CPI-DNN and the calibration of the used Random Forest model for the prediction of the conditional distribution of the variable of interest (G1 and G2).
>
> W4: “The organization of the experimental section makes it quite difficult to read. I would suggest merging Sections 5 and 6 so that a reader does not need to flip back and forth between experimental setup and result.”
>
> R_W4: We merged both sections 5 and 6 in order to make it clearer for the readers.
>
> W5: “Might be worth summarizing the power result from the Supplement in the main paper.“
>
> R_W5: We’ve added in the main paper regarding the power result: “Additional inspection of power (supplement section F) showed that across data generating scenarios, CPI-DNN, Permfit-DNN and conditional-RF showed strong results. Marginal and d0CRT performed only well in scenarios without interaction effects. CPI-RF, cpi-knockoff, LOCO and Lazy VI performed poorly."
>
> W6: “A few of the references are missing identifying information such as arXiv or DOI links, or venue.”
>
> R_W6: We thank the reviewer for raising this point and have repaired the relevant bibliographic records.
>
> Q1: “How good is the normal distribution assumption in practice? It might be interesting or valuable to plot the estimated null distribution in a few different settings with no variable importance.”
>
> R_Q1: An excellent suggestion. In $\textbf{PDF Fig. R2}$, we compared the distribution of the importance scores of a random picked non-significant variable using CPI-DNN and Permfit-DNN through histogram plots and we can emphasize that the normal distribution assumption holds in practice. Also, in $\textbf{PDF Fig. R1}$, we plot the distribution of the p-values provided by CPI-DNN and Permfit-DNN vs the uniform distribution through QQ-plot. We can see that the p-values for CPI-DNN are well calibrated and slightly deviated towards higher values. However, with Permfit-DNN the p-values are not calibrated.

---

> > ### Comment · Reviewer_2imb · 2023-08-14
> >
> > I appreciate the amount of work that the authors have put in responding to my and others' questions. I am quite satisfied with the authors' responses to my questions and the global response, and am happy to increase my score barring any other qualms that come up as I read other reviewer responses.

---

> > > ### Author Response · Authors · 2023-08-17
> > >
> > > We thank the reviewer for the positive appraisal of our revision efforts. This feedback is encouraging us to include the extra plots shared for the revision in the supplementary material to allow future readers to benefit from the improvements obtained through our scientific exchange.

---

### Author Rebuttal · Authors · 2023-08-09

We thank all four reviewers for the time they spent studying our proposed work and for their thoughtful comments. We would also like to thank the reviewers for meaningfully recognizing the strengths of our work while giving us an opportunity to improve it based on their constructive criticism informing about shortcomings and weaknesses.

We have identified a few common points of concern raised by the four reviewers which we will address jointly in what follows. For later reference, our global responses are named G1-G4.

G1

$\textbf{Computational scaling of CPI-DNN and leanness}$:  Following Alg. 1 in the main text and the Random Forest complexity analysis, the complexity can be computed as: $O(n_{estimators} n p^{2} d + B n p)$ where $n_{estimators}$ the number of estimators for Random Forest model, $n$ the number of samples, $p$ the number of variables,$d$ the depth of the trees, $B$ the number of draws, and $mtry=p$ under worst case assumption. Thus, the complexity is linear and quadratic with the number of samples and variables respectively. We are providing a plot summarizing the computation scaling of CPI-DNN as a function of the number of samples and variables ($\textbf{PDF, Fig. R6}$). We can see empirically that CPI-DNN has a linear complexity with the number of samples and a quadratic complexity with the number of variables. As for “computationally lean”, it refers to two facts: (1) there is no need to refit the costly MLP learner to predict y unlike LOCO-DNN (A removal-based method provided with our learner). This can be seen in the $\textbf{PDF Fig. R5}$ where both CPI-DNN and LOCO-DNN achieved a high AUC score and controlled the Type-I error in a highly correlated setting ($n$=$1000$, $p$=50 and $\rho$=$0.8$). However, in terms of computation time, CPI-DNN is far ahead of LOCO-DNN which validates our use of the permutation scheme. (2) The conditional estimation step involved for the conditional permutation procedure is done with an efficient RF estimator, leading to small time difference wrt Permfit-DNN; Overall we obtain the accuracy of LOCO-type procedures for the cost of a basic permutation scheme. Of note, LOCO as used in the main paper, implements the original approach by Williamson et al. 2021.

G2

$\textbf{Random Forest for modeling the conditional distribution and resulting calibration}$: The use of the Random Forest model was to maintain a good non-linear model with time benefits for the prediction of the conditional distribution of the variable of interest. To better frame this idea, in the $\textbf{PDF Fig. R3}$, we are plotting the calibration of the p-values for CPI-DNN (left panel) and the control of type-I error (right panel) as a function of the complexity of the Random Forest (the max depth of the trees). We can see that reducing the depth to 1 or 2, thus making the model overly simple, breaks the control of the type-I errors at the targeted level. With larger depths, the model becomes more conservative. Therefore, the max depth of the Random Forest is chosen based on the performance with 2-fold cross validation. Moreover, the prediction could also “lead to inaccuracies and biased inference if the number of variables becomes too large” as stated in Discussion (section 7) L.311.

G3

$\textbf{Groups needed for very highly correlated settings}$: If we have the case of two highly correlated variables X1 and X2, the corresponding conditional importance of both variables is 0. This problem is linked to the very definition of conditional importance, and not to the CPI procedure itself described in the paper. The only workaround is to eliminate, prior to importance analysis, degenerate cases where conditional importance cannot be defined. This is why, among the future directions, we suggested the use of group-based variable importance where the process is preceded by defining the groups of correlated variables.

G4

$\textbf{Fixes in the plots}$: We’ve fixed the time computation reported in the supplement figures which should be in seconds rather than minutes. Also, for the second row of Fig 3, the results were shifted between the methods. Therefore, we’ve corrected this error in $\textbf{PDF Fig. R7}$. This figure is showing the analysis reported in section 6 (Results).

For convenience, have compiled and enumerated all new results in support of our rebuttal in the PDF attached below. We will refer to the new additional results according to the numbering in the PDF.
For the detailed reviewer-specific responses below, we enumerate to reviewer statements by section, i.e., S for strength, W for weakness, L for limitation and Q for question. Our respective responses are indicated by the prefix “R_”.

---

### Decision · Program_Chairs · 2023-09-21

**Decision:**

Accept (poster)

**Comment:**

Except one reviewer, the scores for this paper are quite good overall. As highlighted by Reviewer XsJL, the discussion of the literature was incomplete and it is surprising that the authors have missed several easy to find papers on this topic. This reviewer seems however to be satisfied by the proposed changes. While the general idea of conditional permutations is not novel, the implementation as done in the paper is original and the experimental comparison is extensive. Despite limited novelty, given the positive scores of all other reviewers, I recommend acceptance. Please make sure however to implement all changes promised during the rebuttal, in particular concerning the discussion of related works.